# HOI-DIFF: TEXT-DRIVEN SYNTHESIS OF 3D HUMAN-OBJECT INTERACTIONS USING DIFFUSION MODELS

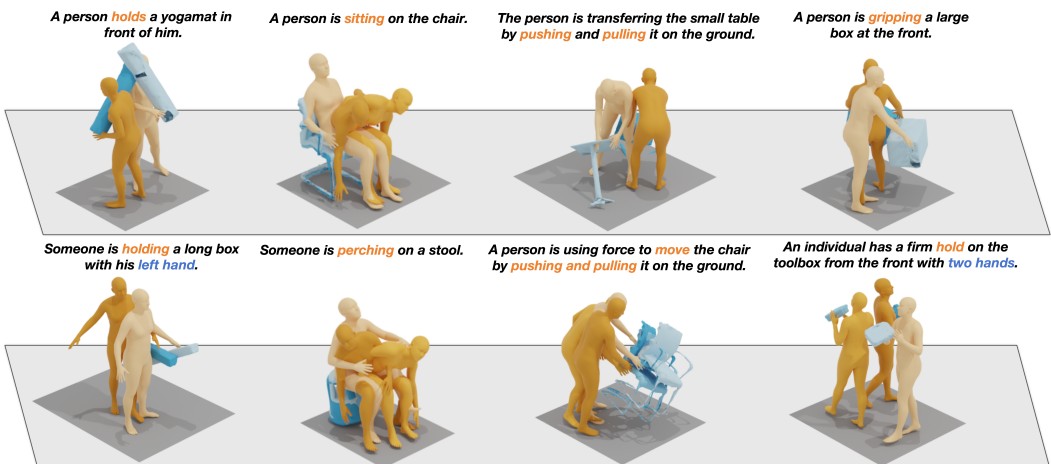

Figure 1: **HOI-Diff generates realistic motions for 3D human-object interactions given a text prompt and object geometry.** Please see the sup. mat. for video results. *Darker color indicates later frames in the sequence. Best viewed in color.*

## ABSTRACT

We address the problem of generating realistic 3D human-object interactions (HOIs) driven by textual prompts. To this end, we take a modular design and decompose the complex task into simpler sub-tasks. We first develop a dual-branch diffusion model (DBDM) to generate both human and object motions conditioned on the input text, and encourage coherent motions by a cross-attention communication module between the human and object motion generation branches. We also develop an affordance prediction diffusion model (APDM) to predict the contacting area between the human and object during the interactions driven by the textual prompt. The APDM is independent of the results by the DBDM and thus can correct potential errors by the latter. Moreover, it stochastically generates the contacting points to diversify the generated motions. Finally, we incorporate the estimated contacting points into the classifier-guidance to achieve accurate and close contact between humans and objects. To train and evaluate our approach, we annotate the BEHAVE dataset with text descriptions. Experimental results on BEHAVE and OMOMO demonstrate that our approach produces realistic HOIs with various interactions and different types of objects. Our code and data annotations will be publicly available.

## 1 INTRODUCTION

Text-driven synthesis of 3D human-object interactions (HOIs) aims to generate motions for both the human and object that form coherent and semantically meaningful interactions. It enables virtual humans to naturally interact with objects, which has a wide range of applications in AR/VR, video games, and filmmaking, etc.

The generation of natural and physically plausible 3D HOIs involves humans interacting with *dynamic* objects in *various* ways according to the text prompts, thereby posing several challenges. First, the variability of object shapes makes it particularly challenging to generate semantically meaningful contact between the human and object to avoid floating objects. Second, the generated HOIs should be faithful to the input text prompts as there are many plausible interactions between human and the same object (*e.g*, a person carries a chair, sits on a chair, pushes or pulls a chair). Text-driven 3D HOI synthesis with a diverse set of interactions is not yet fully addressed. Third, the development and evaluation of 3D HOI synthesis models requires a high-quality human motion dataset with various HOIs and textual descriptions, but existing datasets lack either diverse HOIs (Guo et al., 2022; Plappert et al., 2016; Li et al., 2023a) or detailed textual descriptions with interacting body parts and action (Bhatnagar et al., 2022; Diller & Dai, 2024). It is important to note that CG-HOI (Diller & Dai, 2024) has not made their code or annotations publicly available. In contrast, we will release both our code and annotations.

Current methods cannot fully handle all the challenges. On one hand, recent methods (Kulkarni et al., 2023; Jiang et al., 2022; Hassan et al., 2021; Starke et al., 2019; Zhang et al., 2022b; Wu et al., 2022; Taheri et al., 2022; Pi et al., 2023) can synthesize realistic human motions for HOIs for *static* objects only. They usually synthesize the motion in the last mile of interaction, *i.e*, the motion between the given starting human pose and the final interaction pose, and overlook the movement of the objects when the human is interacting with them. On the other hand, existing methods for motion generation with dynamic objects do not adequately reflect real-world complexity. For instance, they focus on grasping small objects (Ghosh et al., 2023), provide the object motion as conditioning (Li et al., 2023b), predict deterministic interactions between the human and the same object without the diversity (Xu et al., 2023; Razali & Demiris, 2023), consider only a small set of interactions (*e.g*., sit/lift (Kulkarni et al., 2023), sit/lie down (Hassan et al., 2021), sit (Jiang et al., 2022; Zhang et al., 2022b; Pi et al., 2023), grasp (Wu et al., 2022; Taheri et al., 2022)), or investigate a single type of object (*e.g*., chair (Jiang et al., 2022; Zhang et al., 2022b)).

In this paper, we introduce **HOI-Diff** for 3D HOIs synthesis involving humans interacting with different types of objects in diverse ways, which are both physically plausible and semantically faithful to the textual prompt, as shown in Figure 1. Our key insight is to decompose 3D HOIs synthesis into three modules to reduce the complexity of this challenging task. (a) **coarse 3D HOIs generation** that extends the human motion diffusion model (Tevet et al., 2023) to a dual-branch diffusion model (DBDM) to generate both human and object motions conditioning on the input text prompt. To encourage coherent motions, we develop a cross-attention communication module, exchanging information between the human and object motion generation models; (b) **affordance prediction diffusion model** (APDM) that estimates the contacting points between the human and object during the interactions driven by the textual prompt. Our APDM does not rely on the results of the DBDM and thus can recover from its potential errors. Moreover, it stochastically generates the contacting points to diversity the generated motions; and (c) **affordance-guided interaction correction** that incorporates the estimated contacting information and employs the classifier-guidance to achieve accurate and close contact between humans and objects, significantly alleviating the cases of floating objects. Compared with designing a monolithic model, HOI-Diff disentangles motion generation for humans and objects and estimation of their contacting points, which are later integrated to form coherent and diverse HOIs, reducing the complexity and burden for each of the three modules.

For both training and evaluation purposes, we annotate each video sequence in BEHAVE dataset (Bhatnagar et al., 2022) with text descriptions, which mitigates the issue of severe data scarcity for text-driven 3D HOIs generation. In addition, we evaluate our approach on the OMOMO dataset (Li et al., 2023b), which focuses on the manipulation of two hands. Extensive experiments validate the effectiveness and design choices of our approach, particularly for dynamic objects, thereby enabling a set of new applications in human motion generation.

## 2 RELATED WORK

**Human Motion Generation with Diffusion Models.** The denoising diffusion models have been widely used 2D image generations (Rombach et al., 2022; Saharia et al., 2022; Ramesh et al., 2021) and achieved impressive results. Recent work (Zhang et al., 2022a; Tevet et al., 2023; Chen et al., 2023b; Karunratanakul et al., 2023a; Rempe et al., 2023; Ahn et al., 2023; Barquero et al., 2023; Chen et al., 2023a; Dabral et al., 2023; Shafir et al., 2023; Sun & Chowdhary, 2023; Tian et al., 2023; Wei

et al., 2023; Zhang et al., 2023a;b;c; Xie et al., 2023) apply the diffusion model in the task of human motion generation. While these methods have successfully generated human motion, they usually generate isolated motions in the free space without considering the objects the human is interacting with. Our method is primarily focused on motion generation with human-object interactions.

**Scene- and Object-Aware Human Motion Generation.** Recent works condition motion synthesis on scene geometry (Huang et al., 2023; Zhao et al., 2023; Wang et al., 2022a;b). This facilitates the understanding of human-scene interactions. However, the motion fidelity is compromised due to the lack of paired full scene-motion data. Other approaches pKulkarni et al. (2023); Jiang et al. (2022); Hassan et al. (2021); Starke et al. (2019); Zhang et al. (2022b); Pi et al. (2023) instead focus on the interactions with the objects and can produce realistic motions. However, they focus on interacting with static objects with limited interactions. OMOMO (Li et al., 2023b) can generate full-body motion from the object motion. The object motion is needed as input in OMOMO, whereas our method can jointly synthesize human motion and object motion. IMoS (Ghosh et al., 2023) synthesizes the full-body human along with the 3D object motions from textual inputs, but it only focuses on grasping small objects with hands. InterDiff (Xu et al., 2023) predicts whole-body interactions with dynamic objects. Note that the interaction type is deterministic. Different from this, we tackle the motion synthesis task, where the interaction with the same object can be controlled by the text prompt. Recently, there has been a surge of interest in the text-driven synthesis of 3D human-object interactions for dynamic objects, resulting in the development of concurrent works (Diller & Dai, 2024; Wang et al., 2023; Li et al., 2023a; Song et al., 2024; Xu et al., 2024). CG-HOI (Diller & Dai, 2024) and HOIAnimator (Song et al., 2024) uses SMPL parameters as the motion representation, which may result in unsmooth motion due to the potential difficulty in optimization. Instead, we use common skeletal joints similar to most text-to-motion methods, harnessing the power of pre-trained human motion generation models. Chois Li et al. (2023a) relies on the initial state and object waypoints to generate HOIs, which reduces motion diversity for both the human and the object. InterFusion (Dai et al., 2024) and F-HOI (Yang et al., 2024) generate static 3D HOIs from text description, lacking both human and object motions.

**Affordance Estimation.** The affordance estimation on 3D point cloud is studied in Ngyen et al. (2023); Deng et al. (2021); Kokic et al. (2017); Iriondo et al. (2021); Mo et al. (2022); Kim & Sukhatme (2014; 2015). Overall affordance learning is a very challenging task. Instead of predicting the point-wise contact labels, we simplify it by directly regressing the contact points for human-object interactions, making it more tractable without significantly compromising accuracy.

## 3 METHOD

The overview of our proposed approach are illustrated in Figure 2. We introduce a dual-branch Human-Object Interaction Diffusion Model (DBDM), which can produce diverse yet consistent motions, capturing the intricate interplay and mutual interactions between humans and objects (Sec. 3.2). To ensure physically plausible contact between humans and objects, we propose a novel affordance prediction diffusion model (APDM) (Sec. 3.3), whose output will be used as classifier guidance (Sec. 3.4) to correct the interactions at each diffusion step of human/object motion generation.

### 3.1 BACKGROUND

**Motion Representations.** We denote a 3D HOI sequence as $\boldsymbol{x} = \{\boldsymbol{x}^h, \boldsymbol{x}^o\}$. It consists of human motion sequence $\boldsymbol{x}^h \in \mathbb{R}^{L \times D^h}$ and object motion sequence $\boldsymbol{x}^o \in \mathbb{R}^{L \times D^o}$, where $L$ denotes the length of the sequence. For $\boldsymbol{x}^h$, we adopt the redundant representation widely used in human motion generation (Guo et al., 2022) with $D^h = 263$, which include pelvis velocity, local joint positions, velocities and rotations of other joints in the pelvis space, and binary foot-ground contact labels. For the object motion sequence $\boldsymbol{x}^o$, we assume the object geometry is given as an input, and thus we only need to estimate its 6DoF poses in the generation, $i.e$, $D^o = 6$. We represent each object instance as a point cloud of 512 points $\boldsymbol{p} \in \mathbb{R}^{512 \times 3}$.

**Diffusion Model for 3D HOI Generation.** Given a prompt $\boldsymbol{c} = (\boldsymbol{d}, \boldsymbol{p})$, consisting of a textual description $\boldsymbol{d}$ and the object instance's point cloud $\boldsymbol{p}$, a diffusion model $p_\theta(\boldsymbol{x}_{t-1}|\boldsymbol{x}_t, \boldsymbol{c})$[1] learns the reverse diffusion process to generate clean data from a Gaussian noise $\boldsymbol{x}_T$ with $T$ consecutive

---

[1] We use superscripts $h$ and $o$ to denote human and object sequence, respectively. Without a superscript, it means the 3D HOI sequence, containing both $\boldsymbol{x}^h$ and $\boldsymbol{x}^o$. Subscript is used for the diffusion denoising step.

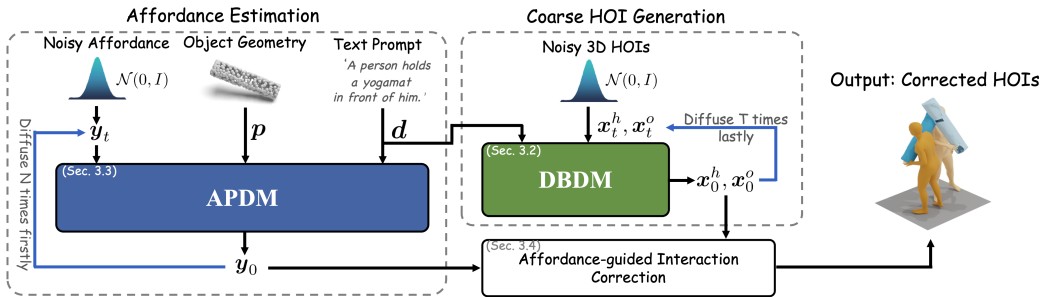

Figure 2: **Overview of HOI-Diff for 3D HOIs generation using diffusion models.** Our key insight is to decompose the generation task into three modules: (a) coarse 3D HOI generation using a dual-branch diffusion model (DBDM), (b) affordance prediction diffusion model (APDM) to estimate the contacting points of humans and objects, and (c) affordance-guided interaction correction, which incorporates the estimated contacting information and employs the classifier-guidance to achieve accurate and close contact between humans and objects to form coherent HOIs.

denoising steps

$$p_\theta(\boldsymbol{x}_{t-1}|\boldsymbol{x}_t, \boldsymbol{c}) := \mathcal{N}(\boldsymbol{x}_{t-1}, \mu_\theta(\boldsymbol{x}_t, t, \boldsymbol{c}), (1 - \alpha_t)\mathbf{I}), \tag{1}$$

where $t$ is the denoising step. Following Tevet et al. (2023), our diffusion model $M_\theta$ with parameters $\theta$ predicts the final clean motion $\boldsymbol{x}_0 = M_\theta(\boldsymbol{x}_t, t, \boldsymbol{c})$. We sample $\mathbf{x}_{t-1} \sim \mathcal{N}(\boldsymbol{\mu}_t, \Sigma_t)$ and compute the mean as in Nichol & Dhariwal (2021)

$$\boldsymbol{\mu}_t = \frac{\sqrt{\alpha_{t-1}}\beta_t}{1 - \alpha_t}\boldsymbol{x}_0 + \frac{\sqrt{1 - \beta_t}(1 - \alpha_{t-1})}{1 - \alpha_t}\boldsymbol{x}_t, \tag{2}$$

where $\alpha_t = \prod_{s=1}^t (1 - \beta_s)$ and $\beta_t \in (0, 1)$ are the variance schedule. $\Sigma_t = \frac{1 - \alpha_{t-1}}{1 - \alpha_t}\beta_t$ (Ho et al., 2020) is a variance scheduler of choice. Similar to $\boldsymbol{x}_t$, $\boldsymbol{\mu}_t$ consists of $\boldsymbol{\mu}_t^h$ and $\boldsymbol{\mu}_t^o$, corresponding to human and object motion, respectively.

Simply adopting the diffusion model described in Eq.(1) would impose a huge burden on the model, which requires joint generation of human and object motion and more critically, enforcement of their intricate interactions to follow the input textual description. In this paper, we propose **HOI-Diff** for 3D HOIs generation, disentangling motion generation for humans and objects and estimation of their contacting points. They are later integrated to form coherent and diverse HOIs, which reduces the complexity and burden for each of the three modules, leading to better generation performance as evidenced by our experiments.

### 3.2 COARSE 3D HOIS GENERATION

First, we introduce a dual-branch diffusion model (DBDM) to generate human and object motions that are roughly coherent. As shown in Figure 3, it consists of two Transformer models (Vaswani et al., 2017), human motion diffusion model (MDM) $M^h$ and object MDM $M^o$, which work similar to Tevet et al. (2023). Specifically, at the diffusion step $t$, they take the text description and noisy motions $\boldsymbol{x}_t^h$ and $\boldsymbol{x}_t^o$ as input and predict clean human and object motions $\boldsymbol{x}_0^h$ and $\boldsymbol{x}_0^o$, respectively.

To enhance the learning of interactions of the human and object when generating their motion, we introduce a Communication Module ($CM$) designed for exchanging feature representations between the human MDM $M^h$ and the object MDM $M^o$. $CM$ is a Transformer block that receives the intermediate feature $\boldsymbol{f}^h, \boldsymbol{f}^o$ from both $M^h$ and $M^o$. It then processes these inputs to generate refined updates based on the cross attention mechanism (Vaswani et al., 2017). The updated feature representations $\tilde{\boldsymbol{f}}_h$ and $\tilde{\boldsymbol{f}}_o$ of the human and object are then conditioned on each other, which are then fed into the subsequent layers of their respective branches to estimate clean human and object motion $\boldsymbol{x}_0^h$ and $\boldsymbol{x}_0^o$, respectively. The $CM$ is inserted at the 4th transformer layer for human MDM and the last layer for object MDM, which was empirically found to work better.

Given the limited data availability for 3D HOI generation, during training, the human motion model $M^h$ finetunes a pretrained human MDM (Tevet et al., 2023). This fine-tuning is critical to ensure the smoothness of the generated human motions. We ablate this design choice in Sec. 4.3. Object

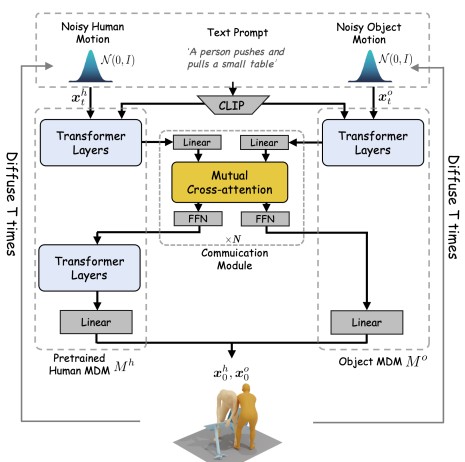
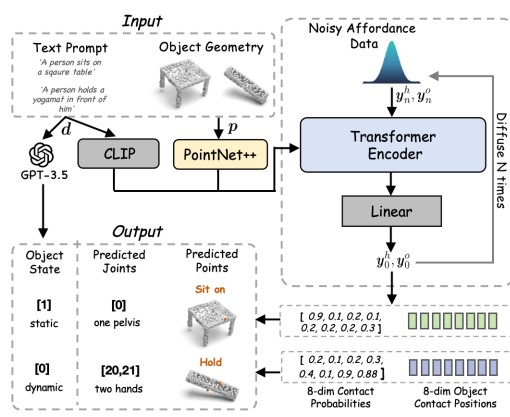

Figure 3: **Illustration of DBDM architecture for coarse 3D HOIs generation.** It has two branches designed for generating human and object motions individually. A mutual cross-attention is introduced to allow information exchange between two branches to generate coherent motions. The human motion model $M^h$ fine-tunes a pretrained MDM (Tevet et al., 2023).

Figure 4: **Illustration of APDM architecture for affordance estimation.** Affordance information of human contact labels, object contact positions, and binary object states are represented together as a noise variable, which is fed into the Transformer encoder to generate clean estimation. The object point cloud and textual prompt are taken as conditional input.

MDM is trained from scratch. We modify the input and output linear layers to take in the object motion which has a different dimension from the human motion. More details of DBDM are in Appendix A.1.

### 3.3 AFFORDANCE ESTIMATION

Due to the complexity of the interactions between a human and object, DBDM alone usually fails to produce physically plausible results, leading to floating objects or penetrations. To improve the generation of intricate interactions, the problem that needs to be solved is to *identify where the contacting areas are* between the human and object. InterDiff (Xu et al., 2023) defines the contacting area based on the distance measurement between the surface of human and object. This approach, however, heavily relies on the quality of the generated human and object motions and cannot recover from errors in the coarse 3D HOI results. In addition, the contact area is diverse even with the same object and interaction type, *e.g*, "sit" can happen on either side of a table. To this end, we introduce an Affordance Prediction Diffusion Model (APDM) for affordance estimation. As illustrated in Figure 4, the input includes a text description $d$ and the object point cloud $p$. Our APDM doesn't rely on the results of the DBDM and thus can recover from the potential errors in DBDM. In addition, it stochastically generates the contacting points to ensure the diversity of the generated motions.

Affordance estimation in 3D point clouds itself is a notably challenging problem (Ngyen et al., 2023; Deng et al., 2021; Kokic et al., 2017; Iriondo et al., 2021; Mo et al., 2022; Kim & Sukhatme, 2014; 2015), especially in the context of 3D HOI generation involving textual prompt. In this paper, we consider eight primary body joints – the `pelvis`, `neck`, `feet`, `shoulders`, and `hands` – as the interacting parts in HOI scenarios. It can effectively model common interactions such as grasping an object with both hands, sitting actions involving the pelvis and back, or lifting with a single hand. We use binary contact labels to determine which joints are in contact with the object. Subsequently, we predict eight corresponding contact points on the object surface, identified as the points closest to the selected body joints. Note that the binary contact label estimation for different body joints are independent, allowing us to handle complex HOIs.

Specifically, at each diffusion time step $n$ of APDM[2], the noisy data consists of human contact labels representing the contact status for the eight primary body joints, denoted as $y_n^h \in \{0,1\}^8$, and the

---

[2]We note that APDM and DBDM work independently. We thus use two symbols to denote the different diffusion time steps to avoid confusion.

eight corresponding contact points on the object surface, denoted as $\boldsymbol{y}_n^o \in \mathbb{R}^{8\times3}$. The model is designed to predict both contact probabilities and contact positions. Subsequently, dynamic selection of contacting body joints is performed by considering predicted probabilities over a specific threshold $\tau$ (set to be 0.6). The corresponding contact points on the object are then determined based on the selected joints. APDM works similar to the diffusion denoising process described in Eq.(1). Besides, we utilize a large language model (ChatGPT) to determine whether the object state $\boldsymbol{y}_0^s \in \{0,1\}$ should be set to static ($\boldsymbol{y}_0^s = 1$) based on the textual description, which can help us better process static objects when synthesizing 3D HOIs, as discussed in the following section. All the clean affordance data is grouped as $\boldsymbol{y}_0 = (\boldsymbol{y}_0^h, \boldsymbol{y}_0^o, \boldsymbol{y}_0^s)$. More implementation details are in Appendix A.2.

## 3.4 Affordance-guided Interaction Correction

With the estimated affordance, we can better align human and object motions to form coherent interactions. To this end, we propose to use the classifier guidance (Dhariwal & Nichol, 2021) to achieve accurate and close contact between humans and objects, significantly alleviating the cases of floating objects.

Specifically, in a nutshell, we define an analytic function $G(\boldsymbol{\mu}_t^h, \boldsymbol{\mu}_t^o, \boldsymbol{y}_0)$ that assesses how closely the generated human joints and object's 6DoF pose align with a desired objective. In our case, it enforces the contact positions of human and object to be close to each other and their motions are smooth temporally. Based on the gradient of $G(\boldsymbol{\mu}_t^h, \boldsymbol{\mu}_t^o, \boldsymbol{y}_0)$, we can perturb the generated human and object motion at each diffusion step $t$ as in Xie et al. (2023); Karunratanakul et al. (2023b),

$$\boldsymbol{\mu}_t^h = \boldsymbol{\mu}_t^h - \tau_1 \Sigma_t \nabla_{\boldsymbol{\mu}_t^h} G(\boldsymbol{\mu}_t^h, \boldsymbol{\mu}_t^o, \boldsymbol{y}_0), \tag{3}$$

$$\boldsymbol{\mu}_t^o = \boldsymbol{\mu}_t^o - \tau_2 \Sigma_t \nabla_{\boldsymbol{\mu}_t^o} G(\boldsymbol{\mu}_t^h, \boldsymbol{\mu}_t^o, \boldsymbol{y}_0). \tag{4}$$

Here $\tau_1$ and $\tau_2$ are different strengths to control the guidance for human and object motion, respectively. Due to the sparseness of object motion features, we assign a larger value to $\tau_2$ compared to $\tau_1$. This applies greater strength to perturb object motion, facilitating feasible corrections for contacting joints. During the denoising stage, to eliminate diffusion models' bias that can suppress the guidance signal, we iteratively perturb $K$ times in the last denoising step. The details are illustrated in Algorithm 1 of Appendix.

How can we define the objective function $G(\boldsymbol{\mu}_t^h, \boldsymbol{\mu}_t^o, \boldsymbol{y}_0)$? We consider three terms here. First, in the generated 3D HOIs, the human and object should be close to each other on the contacting points. We therefore minimize the distance between human contact joints and object contact points

$$G_{con} = \sum_{i \in \{1,2,\dots,8\}} \left\| R\big(\boldsymbol{\mu}_t^h(i)\big) - V\big(\boldsymbol{\mu}_t^o, \boldsymbol{y}_t^o(i)\big) \right\|^2, \tag{5}$$

where $\boldsymbol{\mu}_t^h(i)$ and $\boldsymbol{y}_t^o(i)$ denote the $i$-th available contacting joint indexed by $\boldsymbol{y}_0^h$ and $i$-th object contact point, respectively. $R(\cdot)$ converts the human joint's local positions to global absolute locations, and $V(\cdot)$ obtains the object's contact point sequence from the predicted mean of object pose $\boldsymbol{\mu}_t^o$.

Second, the generated motion of dynamic objects typically follows human movement. However, we observe that when the human interacts with a static object, such as sitting on a chair, the object appears slightly moved. To address this, we immobilize the object's movement in the generated samples if the state is static ($\boldsymbol{y}_0^s = 1$), ensuring that proper contact is established between the human and the static object. The objective is defined as

$$G_{sta} = \boldsymbol{y}_0^s \cdot \sum_{l=1}^{L} \|\boldsymbol{\mu}_t^o(l) - \bar{\boldsymbol{\mu}}_t^o\|^2, \tag{6}$$

where $\boldsymbol{\mu}_t^o(l)$ denotes the object's 6DoF pose in the $l$-th frame. $\bar{\boldsymbol{\mu}}_t^o = \frac{1}{L}\sum_l \boldsymbol{\mu}_t^o(l)$, which is the average of predicted means of the object's pose.

Third, we define a smoothness term $G_{smo}(\mu)$ for the object motion to mitigate motion jittering during contact. Due to the space limit, we explain it in Appendix A.3.

Finally, we combine all these goal functions to as the final objective

$$G = G_{con} + \alpha G_{sta} + \beta G_{smo}, \tag{7}$$

where $\alpha = 500$ and $\beta = 100$ are weights for balance.

## 4 EXPERIMENTS

### 4.1 SETUP

**Dataset.** Since the data designed for studying text-driven 3D HOIs generation is severely scarce, we manually label interaction types, interacting subjects, and contact body parts on top of the BEHAVE dataset (Bhatnagar et al., 2022). We then use GPT-3.5 (OpenAI, 2023) to rephrase and generate three text descriptions for each HOI sequence, increasing the diversity of the data. Specifically, BEHAVE encompasses the interactions of 8 subjects with 20 different objects. It provides the human SMPL-H representation (Loper et al., 2015), the object mesh, as well as its 6DoF pose information in each HOI sequence. To ensure consistency in our approach, we follow the processing method used in HumanML3D (Guo et al., 2022) to extract representations for 22 body joints. All the models are trained to generate $L = 196$ frames in our experiments. In the end, we have 1451 3D HOI sequences along with textual descriptions to train and evaluate our proposed approach. We follow the official train/test split on BEHAVE. We provide more details of the dataset, our annotation process, and annotated textual examples in Appendix I.

In addition, we evaluate our approach on OMOMO dataset (Li et al., 2023b). OMOMO focuses on full-body manipulation with hands. It consists of human-object interaction motion for 15 objects in daily life, with a total duration of approximately 10 hours. It provides text descriptions for each interaction motion. We utilize their object split strategy for both training and evaluation, ensuring the objects between the training and testing sets are different. Additionally, we preprocess human and object motion, similar to our way for the BEHAVE dataset. More details are in Appendix J.

**Evaluation metrics.** We first assess different models for human motion generation using standard metrics as introduced by (Guo et al., 2022), namely *Fréchet Inception Distance (FID)*, *R-Precision*, and *Diversity*. *FID* quantifies the discrepancy between the distributions of actual and generated motions via a pretrained motion encoder. *R-Precision* gauges the relevance between generated motions and their corresponding text prompts. *Diversity* evaluates the range of variation in the generated motions. Additionally, we compute the *Foot Skating Ratio* to measure the proportion of frames exhibiting foot skid over a threshold (2.5 cm) during ground contact (foot height < 5 cm).

To evaluate the effectiveness of HOIs generation, we report the *Contact Distance* metric, which quantitatively measures the proximity between the ground-truth human contact joints and the object contact points. Ideally, we should develop similar metrics, *e.g*, *FID*, to evaluate the *stochastic* HOI generation. However, due to the limited data available in BEHAVE (Bhatnagar et al., 2022), training a motion encoder would produce biased evaluation results. To mitigate this issue, we resort to user studies to quantify the effectiveness of different models. Details will be introduced later.

### 4.2 COMPARISONS WITH EXISTING METHODS

**Baselines.** Our work introduces a novel 3D HOIs generation task not addressed by existing text-to-motion methods, which focus exclusively on human motion generation without accounting for human-object interactions. To compare with existing works, we mainly focus on evaluating human motion generation. We then design different variants of our models for comparing 3D HOIs generation. Specifically, we adopt the prominent text-to-motion methods MDM (Tevet et al., 2023) and PriorMDM* (Shafir et al., 2023) with the following settings. (a) MDM[†]: In this setup, we finetune the original MDM model (Tevet et al., 2023) on the BEHAVE dataset (Bhatnagar et al., 2022) without object motion. (b) MDM*: This variant involves adapting the input and output layers' dimensions of the MDM model (Tevet et al., 2023) to accommodate the input of 3D HOI sequences. This adjustment allows for the simultaneous learning of both human and object motions within a singular, integrated model. (c) PriorMDM* (Shafir et al., 2023): We adapt the ComMDM architecture proposed in Shafir et al. (2023), originally designed for two-person motion generation, to suit our needs for HOIs synthesis by modifying one of its two branches for object motion generation. (d) InterDiff (Xu et al., 2023): While InterDiff is not designed for text-driven synthesis of 3D HOI, we added text conditioning to InterDiff as the baseline. More details are in Appendix C.

**Quantitative Results.** Table 1-left reports the quantitative results on BEHAVE dataset (Bhatnagar et al., 2022). Compared with the baseline methods, our full method achieves the best performance. Specifically, it achieves state-of-the-art results in both *FID*, *R-precision*, and *Diversity*, underscoring its ability to generate high-quality human motions in the context of coherently interacting with objects.

| Method | BEHAVE | | | | | | OMOMO | | | | | |
|---|---|---|---|---|---|---|---|---|---|---|---|---|
| | FID ↓ | R-precision (Top-3) ↑ | Diversity → | Contact Distance ↓ | Pene ↓ | Foot Skate Ratio ↓ | FID ↓ | R-precision (Top-3) ↑ | Diversity → | Contact Distance ↓ | Pene ↓ | Foot Skate Ratio ↓ |
| Real | 0.04 | 0.86 | 12.48 | - | - | - | 0.57 | 0.63 | 9.98 | - | - | - |
| MDM† | 6.77 | 0.34 | 10.81 | - | - | - | 12.28 | 0.23 | 5.56 | - | - | - |
| MDM* | 4.25 | 0.38 | 11.23 | 0.448 | 0.52 | 0.190 | 10.37 | 0.21 | 6.04 | 0.768 | 0.41 | 0.191 |
| PriorMDM* | 4.54 | 0.30 | 10.03 | 0.416 | 0.57 | 0.270 | 9.87 | 0.25 | 6.34 | 0.523 | 0.38 | 0.344 |
| InterDiff | 8.58 | 0.26 | 10.75 | 0.506 | 0.42 | 0.218 | 14.27 | 0.17 | 5.69 | 0.906 | 0.32 | 0.239 |
| Chois | - | - | - | - | - | - | 9.69 | 0.24 | 7.33 | 0.432 | 0.37 | 0.165 |
| **Ours** | **1.62** | **0.46** | **12.02** | **0.347** | 0.51 | **0.182** | **8.76** | **0.31** | **8.13** | **0.326** | 0.39 | **0.141** |

Table 1: **Quantitative results on the BEHAVE and OMOMO dataset.** We compare our method with baselines adapted from existing models. MDM†: fine-tune the original MDM (Tevet et al., 2023) on the BEHAVE dataset without object motion. MDM*: adapting the input and output layers' dimensions of the MDM to accommodate both human and object motions. PriorMDM*: We adapt the ComMDM architecture proposed in Shafir et al. (2023). InterDiff: We add a CLIP encoder in Xu et al. (2023) to support our task. The right arrow → means closer to real data is better. Chois Li et al. (2023a): We remove Initial States & Object waypoints to make a fair comparison.

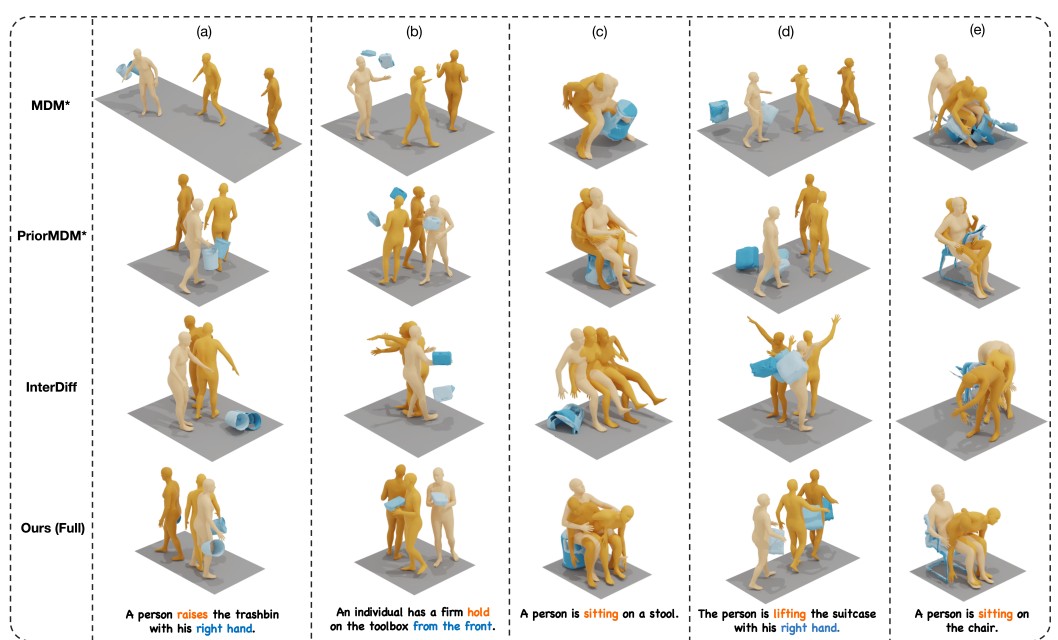

Figure 5: **Qualitative comparisons of our approach and baselines on BEHAVE dataset.** The bottom row, showcasing our method, demonstrates the generation of realistic 3D HOIs with plausible contacts, particularly evident in columns 2 and 4. This contrasts with the baselines, which fail to achieve a similar level of realism and contact plausibility in the interactions. As an additional visual aid, the mesh color gradually darkens over time to represent progression. (Best viewed in color.)

The best *Contact Distance* also suggests that our approach can generate physically plausible HOIs, capturing the intricate interplay interactions between humans and objects.

Table 1-right presents the quantitative results on the OMOMO dataset. We used the train/test split of the OMOMO dataset to evaluate the model's inference capacity on unseen objects, including the small table, white chair, suitcase, and tripod. Our method consistently outperforms other baselines by a considerable margin across all metrics. Notably, due to the distinctiveness of objects in the training and testing sets, the results indicate the effectiveness of our approach in *generalizing to unseen objects*, proving superior performance compared to other models.

**User Study.** The user study requires pairwise comparisons of our method with other baselines on generated interaction quality. The results in Fig. 6 indicate a strong preference for our method: it is favored over the baselines in 89.6% (Ours *vs.* MDM*), 73.8% (Ours *vs.* PriorMDM*) and 95.3% (Ours *vs.* Interdiff). We provide more details in Appendix G

| Variants | BEHAVE | | | | | OMOMO | | | | |
|---|---|---|---|---|---|---|---|---|---|---|
| | FID ↓ | R-precision (Top-3) ↑ | Diversity → | Contact Distance ↓ | Foot Skate Ratio ↓ | FID ↓ | R-precision (Top-3) ↑ | Diversity → | Contact Distance ↓ | Foot Skate Ratio ↓ |
| Real | 0.04 | 0.86 | 12.48 | - | - | 0.57 | 0.63 | 9.98 | - | - |
| *w/o Interaction Correction* | | | | | | | | | | |
| Ours w/o CM | 3.11 | 0.36 | 10.54 | 0.524 | 0.265 | 11.57 | 0.27 | 7.92 | 0.588 | 0.231 |
| Ours w/o pretrain | 2.98 | 0.39 | 11.21 | 0.402 | **0.158** | 10.38 | 0.29 | 7.82 | 0.412 | 0.167 |
| Ours$^{global}$ | 15.37 | 0.28 | 10.85 | 0.375 | 0.274 | 20.22 | 0.21 | 8.02 | 0.366 | 0.348 |
| Ours | 2.10 | 0.38 | 11.26 | 0.415 | 0.205 | 9.12 | 0.29 | 7.97 | 0.397 | 0.193 |
| *w/ Interaction Correction* | | | | | | | | | | |
| Ours w/o $M^o$ & CM | 3.93 | 0.32 | 11.43 | 0.365 | 0.310 | 11.03 | 0.28 | 7.98 | 0.536 | 0.331 |
| Ours $^{joint}$ | 4.37 | 0.31 | 11.25 | 0.421 | 0.342 | 11.52 | 0.27 | 7.92 | 0.547 | 0.325 |
| Ours w/o $G_{con}$ | 2.02 | 0.37 | 11.97 | 0.417 | 0.196 | 9.23 | 0.28 | 8.03 | 0.332 | 0.144 |
| Ours w/o $G_{sta}$ | 1.81 | 0.39 | 11.54 | 0.367 | 0.181 | 9.11 | 0.30 | 8.10 | 0.340 | 0.142 |
| Ours w/o $G_{smo}$ | 1.83 | 0.41 | 11.67 | 0.370 | 0.182 | 8.98 | 0.29 | 8.06 | 0.345 | 0.142 |
| Ours (Full) | **1.62** | **0.46** | **12.02** | **0.347** | 0.182 | **8.76** | **0.31** | **8.14** | **0.326** | **0.141** |

Table 2: **Ablation studies of our model's variants on the BEHAVE and OMOMO datasets.** The right arrow → means closer to real data is better. *w/o CM*: we remove the Communication Module (CM) in the DBDM model. *w/o pretrain*: we train human MDM from scratch on BEAHVE dataset. *global*: we adopt the global human pose representation proposed by Liang et al. (2024) for both the pretraining of human MDM and the finetuning of DBDM. *w/o $M^o$ & CM*: We exclusively finetune the human MDM, while randomly initializing the object motion. Interaction correction is then applied to optimize contact between the human and object. *joint*: We train a single diffusion model that jointly generate human motion, object motion, and affordance. *w/o $G_{con}$/$G_{sta}$/$G_{smo}$*: without contacting/static/smoothness goal function in interaction correction.

**Qualitative Results.** We showcase qualitative comparisons, rendered with SMPL (Loper et al., 2015) shapes, between our approach and the baseline methods in Figure 5. It is observed that the generated HOI motion by other baselines lacks smoothness and realism, where the object may float in the air (*e.g*, the toolbox in Figure 5 (b)). Furthermore, these baseline methods struggle to accurately capture the spatial relationships between humans and objects (*e.g*, the chair in Figure 5 (e)). In stark contrast, our approach excels in creating visually appealing and realistic HOIs. Notably, it adeptly reflects the intricate details outlined in text descriptions, capturing both the nature of the interactive actions and the specific body parts involved (*e.g*, raising the trash bin with the right hand in Figure 5 (a)). For the same object, our method can generate diverse HOIs using different body parts and contact points, as shown in Figure 15 in Appendix.

## 4.3 ABLATION STUDIES

We conduct extensive ablation studies in Table 2 and Figure 7 to validate the effectiveness of different components. We summarize key findings below.

**Object MDM is helpful.** In Table 2, we compare *Ours w/o $M^o$ & CM* and *ours (Full)* to demonstrate the importance of the Object MDM. In *Ours w/o $M^o$ & CM*, we exclusively finetune the human MDM, while randomly initializing the object motion. The Communication Module (CM) is also ignored due to the removed object MDM. Interaction correction is then applied to optimize contact between the human and object. The interaction correction with random initial object motion produces worse results, demonstrating the importance of initial object motion from Object MDM.

**DBDM with Communication Module ($CM$) is critical.** In Table 2, we compare *Ours w/o CM* and *ours* to demonstrate the effectiveness of the Communication Module. When eliminating $CM$, the results drop substantially across all metrics, with a particularly significant decrease in *Contact Distance*. The visual results (w/o $CM$) in Table 7 further validate this point.

**Leveraging the pre-trained Human motion prior can generate better human motions.** We aim to utilize the strong motion prior from the pre-trained human motion model to enhance the realism of the generated motion. Table 2 (*Ours w/o pretrain*) reports the results of training human MDM from scratch, without resuming the weights from the pre-trained MDM (Tevet et al., 2023). Comparing *Ours w/o pretrain* and *Ours* demonstrates the effectiveness of leveraging the pre-trained MDM.

**Interaction Correction makes better HOIs generation.** In Table 2, we compare our full method (*Ours (full)*) to a variant without interaction correction (*Ours*) to demonstrate the effectiveness of interaction correction. The model with interaction correction consistently outperforms the variant across all control accuracy metrics. As shown qualitatively in Figure 7, our full method produces more realistic HOIs with better contact compared to the model without interaction correction. Furthermore,

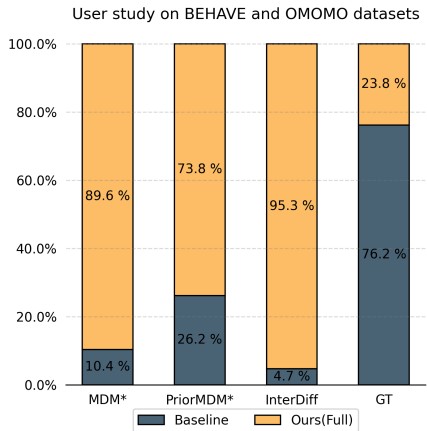

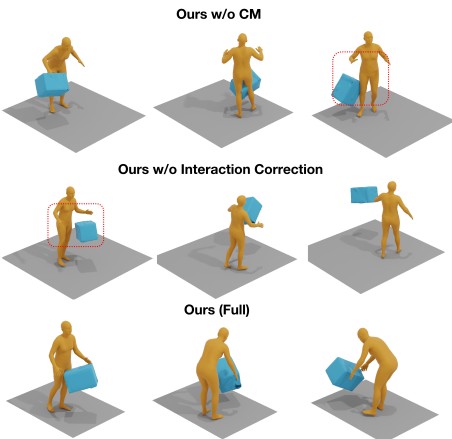

Figure 6: **Perceptual User Study.** Most participants prefer our method over the baselines.

Figure 7: **Visual results of different variants of our model in ablation studies.**

all sub-functions in Interaction Correction contribute to the realistic HOI generation, as demonstrated in *Ours w/o $G_{con}$, w/o $G_{sta}$, w/o $G_{smo}$* of Table 2.

**Why Human MDM and Object MDM are needed separately?** We can ablate this by comparing Table 1 (*MDM\**) and Table 2 (*Ours (w/o Interaction Correction)*. In *MDM\** we jointly learn both human and object motion with a diffusion model. Our superior results demonstrate that separately modeling human motion and object motion with a communication module can achieve better results. A key advantage is that the human motion diffusion model (MDM) can fine-tune a pre-trained MDM (Tevet et al., 2023), leveraging the extensive prior knowledge from the large-scale HumanML3D dataset. In contrast, jointly predicting human and object motion with a single transformer requires training from scratch (due to the change of the model architecture) on the much smaller BEHAVE dataset, which results in poorer human motion results.

**Why not jointly generate motion and affordance with one unified model?** We attempt to generate human motion, object motion, and affordance jointly within the same model, as indicated in the Table 2 (*Ours$^{joint}$*). Our joint prediction concatenates affordance data with motion data along the channel dimension and adjusts the input and output dimensions of MDM to generate motions and affordance simultaneously. Comparing Table 2 *Ours$^{joint}$* and *Ours (full)* demonstrates that our modular design sig-

|  | AP (%) ↑ | L2 Dist ↓ |
|---|---|---|
| Ours $^{joint}$ | 53.67 | 0.384 |
| Ours $^{APDM}$ | **78.54** | **0.272** |

Table 3: APDM evaluation. The reported metrics include Average Precision (AP) for predicted human contact probabilities and L2 Distance (Dist) error for predicted object contact points.

nificantly improves human motion quality, as evidenced by metrics such as FID, R-Precision, and Foot Skate Ratio, as well as the interaction quality measured by Contact Distance. Table 3 further validates that our modular design achieves more accurate affordance estimation, measured by AP and L2 Distance. The improvement is attributed to the fact that affordance learning is highly dependent on the geometry of 3D data and text semantics, rather than human and object motions. Therefore, disentangling these elements enhances their respective performances.

## 5 CONCLUSION

In summary, we presented a novel approach HOI-Diff to generate realistic 3D HOIs driven by textual prompts. By employing a modular design, we effectively decompose the complex task of HOI synthesis into simpler sub-tasks, enhancing the coherence and realism of the generated motions. Our HOI-Diff model successfully generates coarse dynamic human and object motions, while the affordance prediction diffusion model adds precision in predicting contact areas. The integration of estimated affordance data into classifier-guidance further ensures accurate human-object interactions. The promising experimental results on our annotated BEHAVE dataset demonstrate the efficacy of our approach in producing diverse and realistic HOIs.

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

# A  ADDITIONAL DETAILS OF METHODOLOGY

In Sec. 3 of our main paper, we presented the foundational design of each key component in our HOI-Diff pipeline. Here, we delve into an elaborate explanation of model architecture, learning objectives and additional details associated with each crucial component.

## A.1  DUAL-BRANCH DIFFUSION MODEL (DBDM)

The Communication Module (CM) in DBDM is based on the cross attention mechanism. Formally,

$$\tilde{\boldsymbol{f}}^h = \text{MLP}(\text{Attn}(\boldsymbol{f}^h \mathbf{W}_Q, \boldsymbol{f}^o \mathbf{W}_K, \boldsymbol{f}^o \mathbf{W}_V)), \tag{8}$$

$$\tilde{\boldsymbol{f}}^o = \text{MLP}(\text{Attn}(\boldsymbol{f}^o \mathbf{W}_Q, \boldsymbol{f}^h \mathbf{W}_K, \boldsymbol{f}^h \mathbf{W}_V)), \tag{9}$$

where $\text{MLP}(\cdot)$ denotes fully-connected layers, $\text{Attn}(\cdot)$ is the attention block (Vaswani et al., 2017), and $\mathbf{W}_Q, \mathbf{W}_K, \mathbf{W}_V$ are learned projection matrices for query, key, and value, respectively.

The training objective of this full model is based on reconstruction loss

$$\mathcal{L}_{hoi} = \mathbb{E}_{t \sim [1,T]} \| M_\theta(\boldsymbol{x}_t, t, \boldsymbol{c}) - \boldsymbol{x}_0 \|_2^2, \tag{10}$$

where $\boldsymbol{x}_0$ is the ground truth of the HOI sequence.

## A.2  AFFORDANCE PREDICTION DIFFUSION MODEL (APDM).

**Model architecture.** The affordance prediction diffusion model comprises eight Transformer layers for the encoder with a PointNet++ (Qi et al., 2017) to encode the object's point clouds. The training objective of this diffusion model is also based on reconstruction loss

$$\mathcal{L}_{aff} = \mathbb{E}_{t \sim [1,T]} \| A_\theta(\boldsymbol{y}_t, t, \boldsymbol{p}, \boldsymbol{d}) - \boldsymbol{y}_0 \|_2^2, \tag{11}$$

where $\boldsymbol{y}_0$ is the ground-truth affordance data. $\boldsymbol{p}$ and $\boldsymbol{d}$ denote object point cloud and text description (prompt), respectively. $A_\theta$ represents the affordance prediction diffusion model.

**Inferring object state with GPT-3.5-turbo in APDM.** To infer the state of an object, we directly leverage the strong prior knowledge of large language models to derive the result. Specifically, we utilize the GPT-3.5-turbo (OpenAI, 2023) API by inputting specific instructions, allowing it to infer the result directly based on the input HOI text description. The prompt template for instruction is shown in Figure 8.

## A.3  AFFORDANCE-GUIDED INTERACTION CORRECTION.

During the inference stage, it's found that the predicted object contact positions may occasionally be inaccurately positioned, residing either inside or outside the object. To rectify this, we implement post-processing steps that replace these predicted contact points, denoted as $\boldsymbol{y}_0^o$, with their nearest neighbors from the object's point clouds. This adjustment aims to enhance the accuracy of the updated contact points, aligning them more closely with their actual positions on the object's surface. However, employing these updated contact points directly for contact constraints, particularly in the absence of detailed human shape information, introduces a new challenge. It can potentially lead to penetration issues within the contact area while reconstructing the human mesh in the final stage. To mitigate contact penetration, we adopt a method that recalculates points at a specified distance outward, perpendicular to the normal, originating from the object's contact points. This process can formulated as: $\tilde{\boldsymbol{y}}0^o = \hat{\boldsymbol{y}}_0^o + v_n^i * d$, where $i \in \{1, 2\}$ indicates the $i^{th}$ object contact points, $v_n^i$ denotes the normal vector at that point and $d = 0.05$ is a contact distance threshold.

As for smoothness term, we formulate it as

$$G_{smo} = \sum_{l=1}^{L-1} \| \boldsymbol{x}_0^o(l+1) - \boldsymbol{x}_0^o(l) \|^2, \tag{12}$$

where $\boldsymbol{x}_0^o(l)$ is the predicted 6DoF pose of the object in the $l$-th frame.

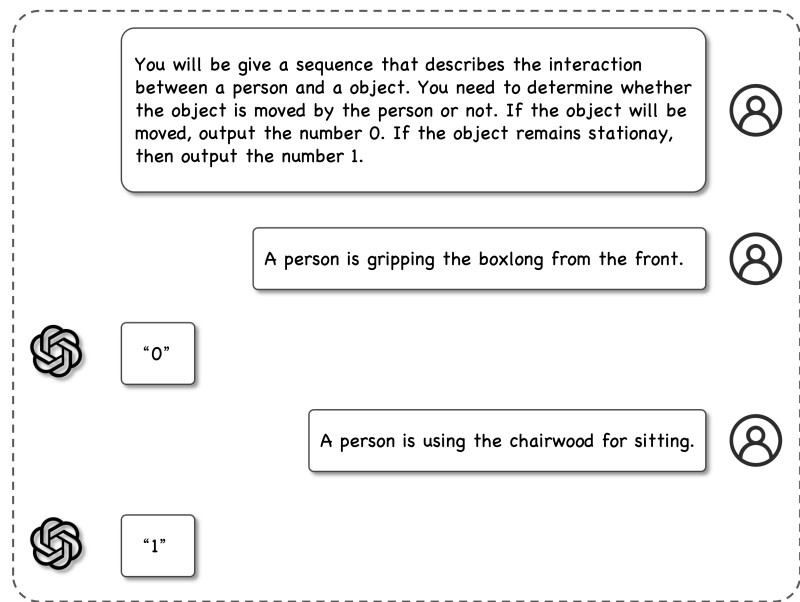

Figure 8: **Prompt template for inferring object state.**

---

**Algorithm 1** Affordance-guided Interaction Correction
---
**Require:** Input $c = (d, p)$ consisting of a textual description $d$ and object point cloud $p$, HOI-Diff model $\mathbf{M}_\theta$, objective function $G(\mu_t^h, \mu_t^o, y_0)$, and estimated affordance $y_0 = (y_0^h, y_0^o, y_0^s)$.
1:  $x_T^h, x_T^o \leftarrow$ sample from $\mathcal{N}(\mathbf{0}, \mathbf{I})$
2:  $K = 1$
3:  **for all** $t$ from $T$ to 1 **do**
4:      $x_0^h, x_0^o \leftarrow M_\theta(x_t^h, x_t^o, t, c)$ # Get $\mu_t^h, \mu_t^o$ according to Eq.(2) with $\Sigma_t$
5:      **if** $t = 1$ **then**
6:          $K = 100$
7:      **end if**
8:      **for all** $k$ from $K$ to 1 **do** # Separately perturb
9:          $\mu_t^h \leftarrow \mu_t^h - \tau_1 \Sigma_t \nabla_{\mu_t^h} G(\mu_t^h, \mu_t^o, y_0), \quad \mu_t^o \leftarrow \mu_t^o - \tau_2 \Sigma_t \nabla_{\mu_t^o} G(\mu_t^h, \mu_t^o, y_0)$
10:     **end for**
11:     $\mathbf{x}_{t-1}^h \sim \mathcal{N}(\mu_t^h, \Sigma_t), \quad \mathbf{x}_{t-1}^o \sim \mathcal{N}(\mu_t^o, \Sigma_t)$
12: **end for**
13: **return** $x_0^h, x_0^o$

---

# B    IMPLEMENTATION DETAILS

Both our DBDM and APDM are built on the Transformer (Vaswani et al., 2017) architecture. Similar to MDM (Tevet et al., 2023), we employ the CLIP model to encode text prompts, adhering to a classifier-free generation process. Our models are trained using PyTorch (Paszke et al., 2019) on 1 NVIDIA A5000 GPU. We set control strength of guidance as $\tau_1 = 1$, $\tau_2 = 100$, and $\Sigma_t = min(\Sigma_t, 0.01)$. Both the DBDM and APDM are trained on the same data for 20k steps.

Both the DBDM and APDM architectures of HOI-Diff are based on Transformers with 4 attention heads, a latent dimension of 512, a dropout of 0.1, a feed-forward size of 1024, and the GeLU activation (Hendrycks & Gimpel, 2016). The number of learned parameters for each model is stated in Table 4.

Our training setting involves 20k iterations for the DBDM and 10k iterations for the APDM model. These iterations utilize a batch size of 32 and employ the AdamW optimizer (Loshchilov & Hutter, 2017) with a learning rate set at $10^{-4}$. We use $T$=1000 and $N$=500 diffusion steps in DBDM and APDM, respectively.

## C ADDITIONAL DETAILS OF BASELINES

- MDM$^{finetuned}$: We finetune MDM (Tevet et al., 2023) on BEHAVE dataset without considering the object motion.

- MDM*: We extend the original feature dimensions of the input and output processing in MDM (Tevet et al., 2023) from $D^h$ to $D^h + D^o$, enabling support for HOIs sequences. The model is trained from scratch on BEHAVE dataset (Bhatnagar et al., 2022).

- PriorMDM*: The proposed approach for dual-person motion generation employs paired fixed MDMs (Tevet et al., 2023) per individual to ensure uniformity within generated human motion distributions. This design leverages a singular ComMDM to coordinate between the two branches of fixed MDM instances, streamlining training and maintaining consistency across generated motions. Given that both branches are based on MDM that pretrained on human motion datasets, direct utilization of them for human-object interactions in our task is infeasible. We maintain one branch dedicated to humans, leveraging pre-trained weights, while adapting the input and output processing of another branch specifically for generating object motion. Following this, we fine-tune the human MDM branch while initiating the learning of object motion from scratch within the object branch. Eventually, we integrate ComMDM to facilitate communication and coordination between these distinct branches handling human and object interactions.

- InterDiff: InterDiff (Xu et al., 2023) is originally designed for a prediction task rather than text-driven HOIs generation. To tailor it to our task, we replace its Transformer encoder with a CLIP encoder and modify its feature dimensions of the input and output layers.

- Chois: Chois (Li et al., 2023a) is a work closely related to ours. For a fair comparison, we remove the initial states of the human and object, exclude object waypoints as conditions, and adopt the same motion representation as input.

To ensure fair comparisons, all the above baselines as well as our own models are all trained on BEHAVE and OMOMO datasets for 20k steps.

## D ADDITIONAL DETAILS OF EVALUATION METRICS

For detailed information regarding metrics employed in human motion generation, including *FID*, *R-Precision*, and *Diversity*, we refer readers to Tevet et al. (2023); Guo et al. (2022) for comprehensive understanding.

**Contact Distance.** Expanding on the concept of *Contact Distance*, we utilize the *chamfer distance* metric to quantify the closeness between human body joints and the object surface. This computation leverages ground-truth affordance data that includes human contact labels and object contact points,

$$ContactDistance = \frac{1}{L} \sum_{l}^{L} CD(\hat{\boldsymbol{x}}_l^h, \hat{\boldsymbol{p}}_l), \tag{13}$$

where $\hat{\boldsymbol{x}}_l^h$ represents two human contact joints at the $l$-th frame, indexed according to ground-truth contact labels. Additionally, $\hat{\boldsymbol{p}}_l$ denotes two object contact points derived from the object motion $\boldsymbol{x}_l^o$ at frame $l$, also indexed based on ground-truth information. $CD$ denotes the *chamfer distance*.

**Penetration Score.** We followed the Li et al. (2023a) to compute the penetration score (Pene), each vertex of the body ($V_i$) is queried against the precomputed Signed Distance Field (SDF) of the object. This process yields a corresponding distance value for each vertex. The penetration score is then formalized as:

$$Pene = \frac{1}{n} \sum_{i=1}^{n} |min(d_i, 0)|, \tag{14}$$

measured in centimeters (cm).

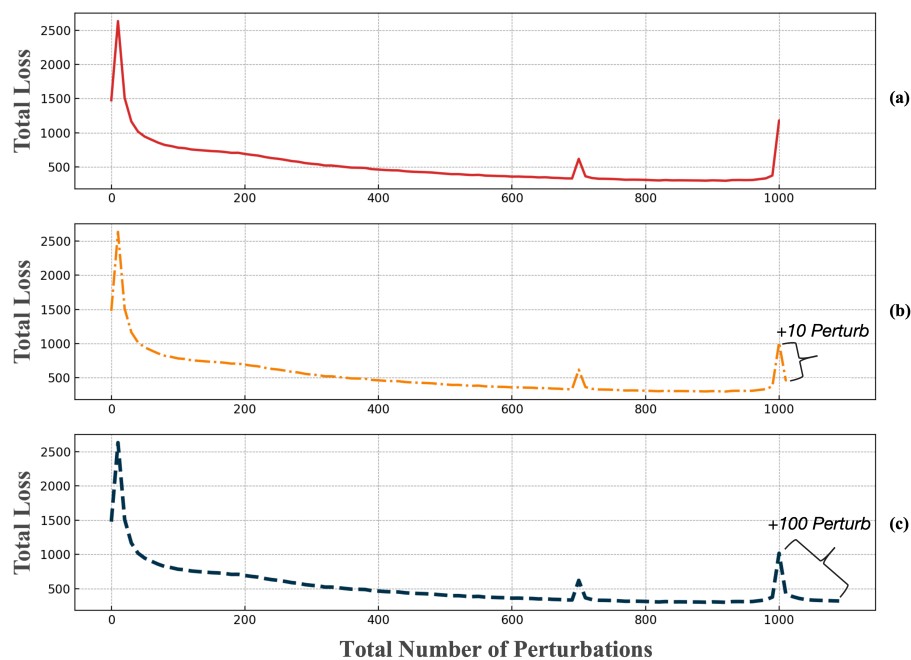

Figure 9: **Effect of different total numbers of perturbations in the whole denoising process.** (a) Perturb one time in each denoising step (in total $T = 1000$). (b) Perturb one time in first $T - 1$ denoising steps, and repeatedly perturb 10 times in the final denoising step. (c) Perturb one time in first $T - 1$ denoising steps, and repeatedly perturb 100 times in the final denoising step.

| Model | DBDM | APDM |
|---|---|---|
| Parameters ($\cdot 10^6$) | 8.82 | 38.92 |

Table 4: **Model Parameters.** The number of learned parameters of our two core architectures.

| Method | MDM* | PriorMDM* | Ours (Full) |
|---|---|---|---|
| Time (s) | 32.3 | 38.6 | 118.0 |
| Component | APDM | DBDM | Interaction Correction |
| Time (s) | 24.2 | 46.4 | 47.4 |

Table 5: **Inference Time (on NVIDIA A5000 GPU).** We report the inference time for baselines, our full method, and its key components.

## E    INFERENCE TIME

In Table 5, we provide the inference times for both baselines and our full method, including its key components. All measurements were conducted using an NVIDIA A5000 GPU. Training an additional model for affordance information and using classifier guidance for interaction correction do contribute to increased inference costs. However, despite the longer inference time, our complete method notably enhances the accuracy of 3D HOIs generation.

| | Params (M) | FID ↓ | R-precision (Top-3) ↑ |
|---|---|---|---|
| MDM* | 49.85 | 6.98 | 0.36 |
| Ours (Full) | 47.74 | 1.62 | 0.46 |

Table 6: With comparable model size, the performance results of MDM* and Ours (Full).

## F    ADDITIONAL ABLATION STUDIES

**Different perturbing times in classifier guidance.** As discussed in Sec. 3.4, in the later stage of classifier guidance, diffusion models tend to strongly attenuate the introduced signals. Therefore, we

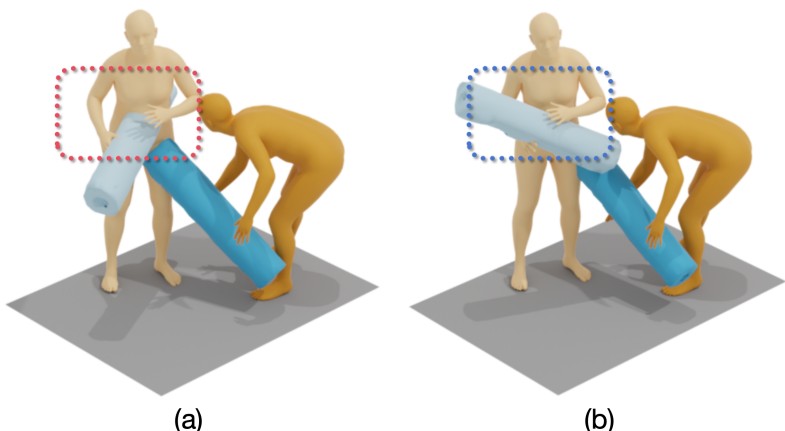

(a) (b)

Figure 10: **Effect of different control strengths for classifier guidance.** (a) We use equal strengths of $\tau_1 = 1, \tau_2 = 1$ to perturb the predicted mean of human motion and object motion, respectively. (b) We use different strengths of $\tau_1 = 1, \tau_2 = 100$ for the perturbation. We can see that different strengths work better.

iteratively perturb the predicted mean of motion for $K$ times at the final denoising step. In Figure 9, we present the ablation results, illustrating the impact of different numbers of perturbations. Notably, we observe that employing 100 perturbations leads to re-convergence and yields the desired results.

**Different guidance strength.** As detailed in Sec. 3.4, we employ distinct control strengths for classifier guidance, considering the varying feature densities in predicted human and object motion. Rather than employing equal control strengths, we opt to assign a higher control strength to object motion, allowing it to closely align with human contact joints, as illustrated in Figure 10.

**Different model with comparable model size.** Although our method involves a slightly larger number of model parameters, our model is specifically designed for HOI generation. As seen in the Table 6, if we attempt to scale MDM* to the same model size, its performance remains subpar.

# G  USER STUDY

For each method, we select 15 prompts from the BEHAVE dataset and 10 prompts from the OMOMO dataset, covering various interaction types and object items. We sample twice with each prompt to gather a total of 50 results. 40 participants are asked to choose their most preferred generation results from these samples. This user study requires pairwise comparisons of our method with other baseline on generated interaction quality, as shown in Figure 11.

# H  ADDITIONAL QUALITATIVE RESULTS

In this section, we present additional qualitative results showcasing the model's performance evaluated on the OMOMO dataset, and the effectiveness of APDM.

**Qualitative results on OMOMO dataset.** We present additional qualitative results on the OMOMO dataset, rendered with SMPL (Loper et al., 2015) shapes, as shown in Figure 12. It is evident that our method can generalizes effectively to unseen objects and produce realistic 3D human-object interactions.

**Qualitative results of APDM.** To verify the accuracy of estimated contact points on object surface, we provide additional visual results in Figure 14. It can be seen that our method can predict realistic and practical contact points based on text descriptions. With APDM, we even can generate different interactions with the same object based on the input description, as shown in the Figure 15.

## Which one looks more realistic and is coherent with text description?

Prompt: *The person is gripping with suitcase with his right hand.*

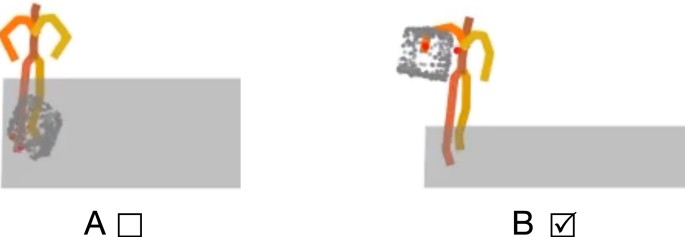

A ☐                    B ☑

Figure 11: **An example question for our text-to-hoi user study.**

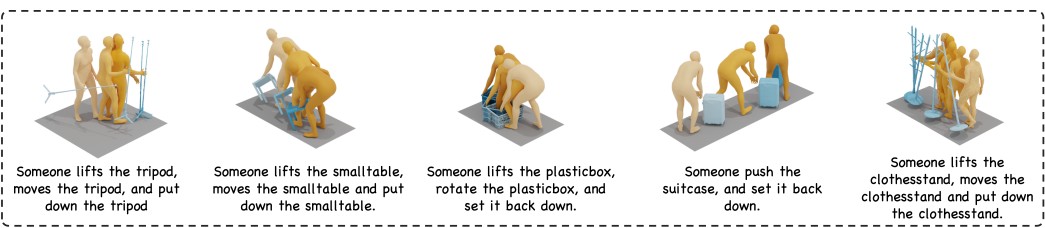

Someone lifts the tripod, moves the tripod, and put down the tripod

Someone lifts the smalltable, moves the smalltable and put down the smalltable.

Someone lifts the plasticbox, rotate the plasticbox, and set it back down.

Someone push the suitcase, and set it back down.

Someone lifts the clothesstand, moves the clothesstand and put down the clothesstand.

Figure 12: Additional qualitative evaluation on OMOMO dataset. Given object geometry and text description, our method can generate high-quality human-object interactions even for the unseen objects (tripod, smalltable, suitcase).

**Generalization capability.** To verify the model's generalization capability, except of unseen object test on OMOMO dataset, we also downloaded several objects from Sketchfab[3], adjusted them to a reasonable scale, and used them as inputs. As shown in Figure 13, our model successfully establishes reasonable HOI contact with these previously unseen objects.

## I   ANNOTATION FOR BEHAVE DATASET

**Text Annotating Process.** Initially, we manually annotate the interaction types and the specific human body parts involved, delineating actions like "lift" associated with the "left hand" or "hold" involving "two hands". Subsequently, to generate complete sentences, we leverage the capabilities of GPT-3.5 to assist in formulating the entirety of the description.

**Examples of Annotated Textual Descriptions.** In Table 7, we showcase a selection of our annotated textual descriptions for the BEHAVE dataset (Bhatnagar et al., 2022).

**Analysis of Annotated Textual Descriptions.** All text descriptions encompass 36 distinct interaction verbs associated with 20 different objects. Figure 16 illustrates the frequency of each verb, indicating their respective occurrences.

**Affordance Data.** Our affordance data includes 8-dimensional human contact labels and object contact points. We employ *chamfer distance* to measure the distance between all human body joints and object surface points. Following a predefined distance threshold $\gamma = 0.12$, we identify the 8

---

[3]https://sketchfab.com/

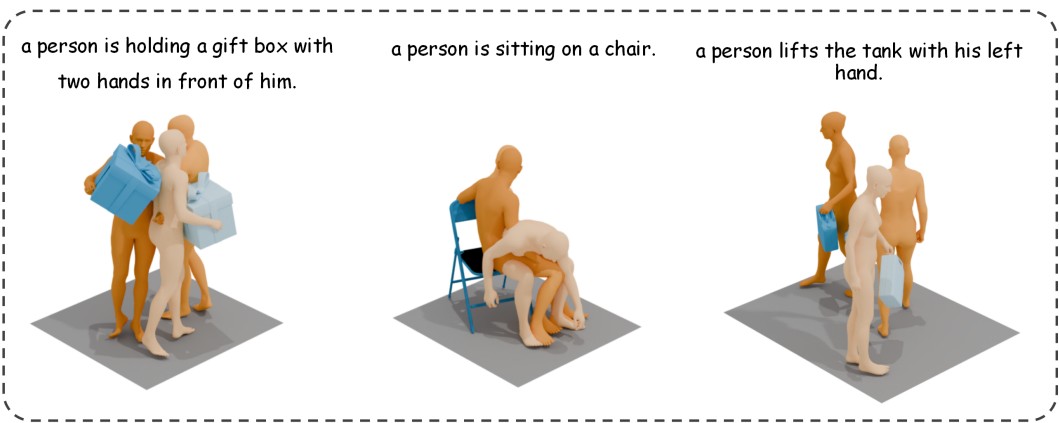

Figure 13: Additional qualitative evaluation on unseen objects.

| Object | Textual Descriptions |
|---|---|
| *backpack* | A person is carrying the backpack in front. |
| | The person is raising a backpack with his right hand. |
| | The person at the front presently has control over the backpack. |
| *chairwood* (*wooden chair*) | A person is using the chairwood for sitting. |
| | The person is propelling the chairwood on the ground. |
| | Someone is hoisting a chairwood by his left hand. |
| *tablesquare* (*square table*) | A person is lifting the tablesquare, utilizing his left hand. |
| | Someone is clutching onto a tablesquare from the front. |
| | An individual is moving the tablesquare back and forth. |
| *boxlong* (*long box*) | A person is gripping the boxlong from the front. |
| | A person is raising the boxlong using his left hand. |
| | Someone hoists the boxlong with his left hand. |
| *toolbox* | Someone is grasping the toolbox upfront. |
| | The person has a firm hold on the toolbox with his right hand. |
| | A person is gripping the toolbox with his left hand. |
| *yogaball* | A person is shifting a yogaball back and forth on the floor using his hands. |
| | The person is occupying a yogaball. |
| | A person is employing an yogaball to engage in an upper body game. |

Table 7: **Examples of our annotated textual descriptions for the BEHAVE dataset rephrased by GPT-3.5 (OpenAI, 2023).**

contact points on the object surface corresponding to the 8 primary human body joints. Subsequently, we derive the human contact labels by encoding the indexes of contact joints into an 8-dimensional vector represented by binary values.

## J    ADDITIONAL DETAILS OF OMOMO DATASET

The OMOMO dataset comprises data captured for a total of 15 objects. Adhering to their official split strategy depicted in Li et al. (2023b)(Figure 5), we allocate 10 objects for training and 5 objects for testing. This split allows us to further evaluate the model's generalization ability to new objects. Notably, the OMOMO dataset itself provides text annotation, and we use GPT-3.5 to add subjects

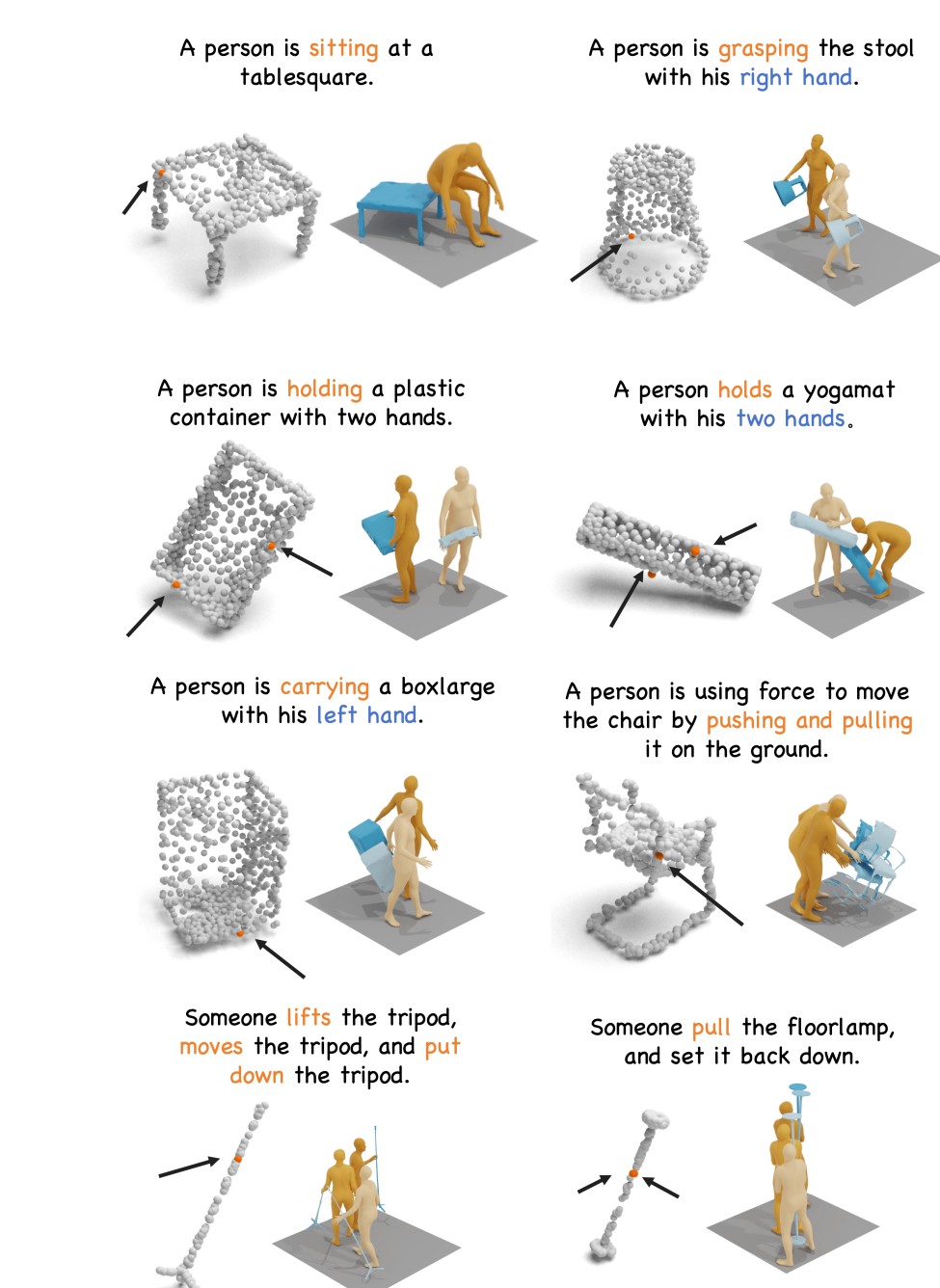

Figure 14: Visual results of estimated contact points. Our APDM, trained on the BEHAVE dataset, can accurately estimating contact positions for objects based on textual descriptions. Furthermore, it showcases the capability to generalize to unseen objects in the OMOMO dataset, as demonstrated in the last row.

to it and embellish it appropriately. For affordance data, we preprocess it the same way we handle BEHAVE.

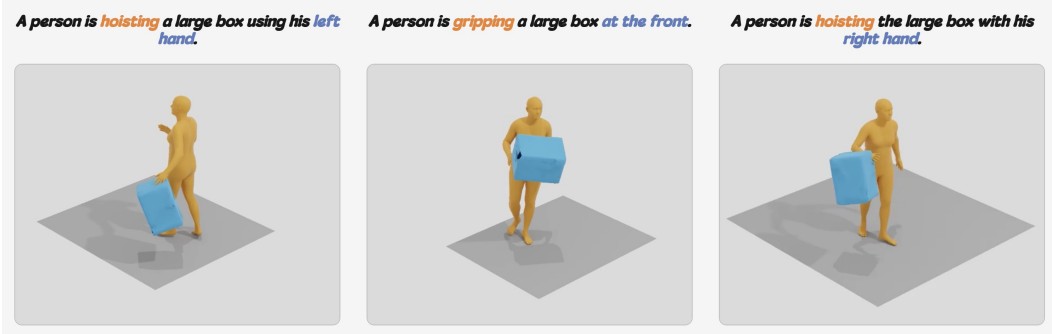

Figure 15: Leveraging the power of the APDM module, our method can generate diverse HOIs for the same object using different contacting body parts and contact points.

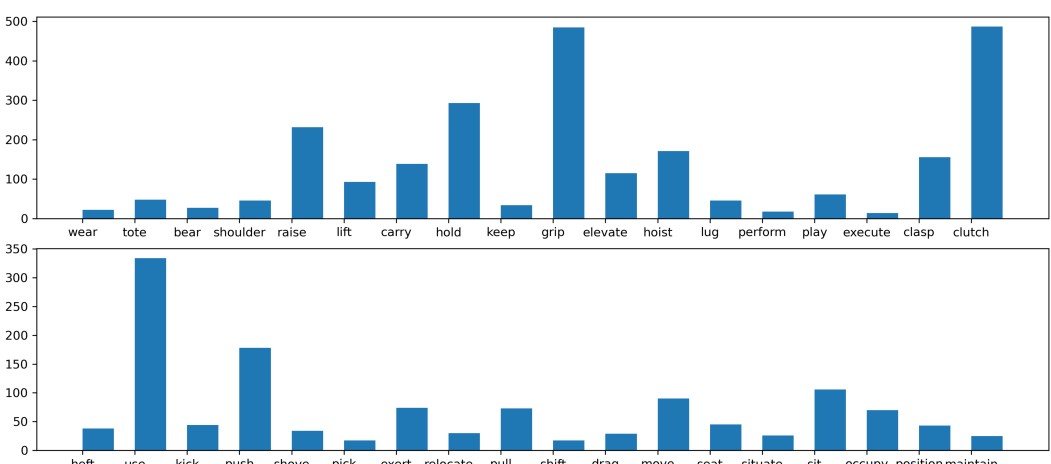

Figure 16: **Analysis of word frequency** We count the occurrences of each interaction verb from all text descriptions to illustrate their respective frequencies.

## K COMMON QUESTIONS

**Why use Skeletal Pose Representation rather than SMPL parameters?** Most state-of-the-art text-to-motion methods adopt the skeletal pose representation proposed by Guo et al. (2022), demonstrating excellent performance and stability. While some works (Azadi et al., 2023) argue that SMPL parameters (Loper et al., 2015) contains shape and global information, it does not generate as smooth motions as skeletal-based approaches. Consequently, we adopt the skeletal pose representation and aim to leverage strong pose priors from the pretrained text-to-motion model (Tevet et al., 2023) to ensure the authenticity of generated human motion.

**Can we handle multi-phase interactions between humans and objects?** Due to the lack of fine-grained textural descriptions in the current 3D HOI dataset, we primarily consider only one interaction phase. However, we have found that an LLM can still reason well for multiple phases given a template such as: *You will be given a sentence that describes an interaction between a person and an object across multiple phases. Your task is to divide the interaction into phases based on the state of the object and determine the state for each phase. If the object is being moved by the person during a phase, output the number 0. If the object remains stationary during a phase, output the number 1.*

For example, given the text description: *The box is on the ground. A person is picking up the box and holding it forward, then putting the box towards the table. The box is on the table" The result from GPT-3.5-turbo: "Phase 1: The box is on the ground - State: 1 (stationary); Phase 2: The person is picking up the box and holding it forward - State: 0 (moved); Phase 3: The person is putting the box*

*towards the table - State: 0 (moved); Phase 4: The box is on the table - State: 1 (stationary).* We will address the generation of multiple phases of 3D HOI in future work.

**Can we generate hand motion with articulated fingers?** The BEHAVE and OMOMO datasets do not capture and provide raw hand parameters, despite utilizing SMPLH and SMPLX models to fit human body meshes for rendering. Consequently, in this paper, we focus solely on whole-body human motion, excluding articulated hand and finger movements.

**Why do we use large language models (LLMs) to predict object state based on the input description?** We aim to leverage LLMs for inferring object states, and our results demonstrate that they perform efficiently and effectively. As shown in the Table 8, we evaluated the performance of object state prediction with GPT-3.5-turbo (OpenAI, 2023) and obtained an average precision of 95.6% on the validation set, with an average response time of 0.518 seconds. The results suggest that GPT-3.5-turbo is sufficiently accurate without adding significant overhead. We also evaluated the prediction performance using other LLMs, including Gemini-1.5-Pro-Exp-0801 (Reid et al., 2024) (99.4%, 0.259s), Gemma-2-27B (Team et al., 2024) (98.6%, 0.522s), and LLaMA-2-13B (Touvron et al., 2023) (94.4%, 0.521s), the latter two being publicly available.

To further validate the effectiveness of the LLM module, we modified the APDM module by adding an MLP head to predict the object status. The newly added MLP takes in the features consisting of object geometry information and CLIP embeddings. We used an MSE loss. We got average precision 79.5% and average time 2.42s for this design on the validation set, which is significantly worse than the results of GPT-3.5-turbo (95.6%, 0.518s), Gemma-2-27b (98.6%, 0.522s), Gemini-1.5-Pro-Exp-0801 (99.4%, 0.259s) and LLaMA-2-13B (4.4%, 0.521s).

|  | Acc (%) ↑ | Time (s) ↓ |
|---|---|---|
| GPT-3.5 | 95.6 | 0.518 |
| Gemini-1.5-Pro-Exp-0801 | 99.4 | 0.259 |
| Gemma-2-27B | 98.6 | 0.522 |
| LLaMA-2-13B | 99.4 | 0.259 |
| APDM + MLP | 79.5 | 2.420 |

Table 8: LLMs' inference accuracy (Acc) and average inference time (Time) on object state prediction.

In future work, we believe the LLM can play a more important role in 3D HOI, e.g. providing high-level instruction for more complex human-object interactions, and our initial use of the LLM offers insights into its potential applications and how it can be effectively utilized.

## L  SUPPLEMENTARY VIDEO

Beyond the qualitative results presented in the main paper, our supplementary materials offer comprehensive demos that provide an in-depth visualization of our task, further showcasing the effectiveness of our approach.

In these demonstrations, we highlight the better performance of our method, HOI-Diff, in producing diverse and realistic 3D HOIs while maintaining adherence to physical validity. Notably, the visualizations show that HOI-Diff consistently generates smooth, vivid interactions, accurately capturing human-object contacts.

Additionally, we present the visual ablation results and emphasize the significance and effectiveness of our affordance-guided interaction correction, underscoring its substantial impact on improving the overall performance and quality of the generated 3D HOIs.

## M  LIMITATIONS

The existing datasets for 3D HOIs are limited in terms of action and motion diversity, posing a challenge for synthesizing long-term interactions in our task. Furthermore, the effectiveness of our model's interaction correction component is contingent on the precision of affordance estimation. Despite simplifying this task, achieving accurate affordance estimation remains a significant challenge, impacting the overall performance of our model. A promising direction for future research involves integrating a sophisticated affordance model pre-trained on an extensive 3D object dataset, along with text prompts. Such an advancement could significantly enhance the realism and accuracy of human-object contact in our model, leading to more natural and precise HOIs synthesis.

# N    SOCIAL IMPACTS

On the positive side, it may offers the research community valuable insights into understanding human behaviors. On the negative side, it remains uncertain whether individuals can be identified solely based on their poses and movements. However, compared to traditional input images of people, this method poses a lower risk of invading personal privacy.

