# OpenReview forum: "HOI-Diff: Text-Driven Synthesis of 3D Human-Object Interactions using Diffusion Models"
_ICLR.cc/2025/Conference — Submitted to ICLR 2025_

### Official Review · Reviewer_EXrp · 2024-10-29

**Soundness:** 4
**Presentation:** 3
**Contribution:** 3
**Rating:** 6
**Confidence:** 4

**Summary:**

This paper introduces a novel approach for generating realistic 3D human-object interactions(HOIs) based on textual prompts and object geometry. The authors propose a modular design that decomposes the task into simpler sub-tasks: a dual-branch diffusion model(DBDM) for generating human and object motions, an affordance prediction diffusion model(APDM) for estimating contact information, and classifier-guidance mechanism for correcting interactions to ensure accurate contacts. Additionally, the authors annotate the BEHAVE dataset with text labels to enable the training of models. The method is evaluated on both the BEHAVE and OMOMO datasets, with results showing the ability to produce realistic and diverse HOIs with different types of objects.

**Strengths:**

1.	The novel decomposition of the complex task into sub-tasks allows for specialized models to handle different aspects of the problem, and the integration of them brings better performance in generating HOIs.
2.	The authors manually annotate the BEHAVE dataset with text label, which extends the HOI dataset with language modality and enables downstream tasks such as text-to-hoi generation, motion captioning, etc.
3.	The authors conduct various experiment to validate the effectiveness of the method, and also perform generalization test on the unseen OMOMO dataset.
4.	The paper is well-written and easy to understand.

**Weaknesses:**

1.	The R-precision, FID and Diversity metric only evaluate the human motion with the text prompt, rather than the whole generated HOI, which cannot comprehensively measure the quality of the generated HOI. The Contact Distance alone only measures the quality of generated contact. Is there any additional metric that could assess the full HOI? For example, whether the human penetrates with objects could be evaluated by Interpenetration Ratio metric. I hope the authors could provide more effective metrics and perform some evaluation on them.
2.	Also, the R-precision value is far too low in comparison with the ground truth(and also other T2M works, for example, MLD achieves 0.796 in R-Precision@3 in HML3D test set), which may indicate the inconsistency between text prompt and generated HOI. Would you please explain why this occurs and some possible solutions? And I’d like to see some comparisons with more recent Text2HOI works such as Controllable HOI(Jiaman Li, et al).

**Questions:**

1.	What’s the feature extractor you used in the evaluation? Did you re-train them or just use the original feature extractor (Guo et al.,  2022)?
2.	Is the affordance in APDM estimated for the whole HOI sequence or for every single frame?
3.	In the affordance estimation stage, you only consider eight body joints for contact. Is this enough for modeling the HOI process? For example, one may kneel on the yoga ball with his knees, which are not included.
4.	Does the model generate constant motion length (as mentioned in L335)?

---

> ### Author Response · Authors · 2024-11-21
> **Rebuttal by Authors**
>
> We thank Reviewer Exrp for your comments and constructive suggestions. We respond below to your comments and questions.
>
> **W1: Is there any additional metric that could assess the full HOI?**
>
> **A1:**  We thank Reviewer EXpr for the valuable suggestion to include the penetration score in our evaluation. We have updated Table 1 in the revised PDF to include the penetration metrics, highlighted in blue. Please refer to FAQ2 for more details. We believe that the penetration score, alongside R-precision, FID, Diversity, and contact distance, provides a comprehensive evaluation of the quality of 3D HOI.
>
> **W2.1: the R-precision value is far too low in comparison with the ground truth.**
>
> **A2.1:** The primary reason likely lies in the relatively small scale of the BEHAVE dataset. We fine-tuned the evaluator on this dataset because the original evaluator, trained on the HumanML3D dataset, did not generalize well to Behave. Consequently, the fine-tuned evaluator may exhibit slight overfitting to the ground truth data in Behave, which could explain the comparatively lower R-precision of predictions (both ours and the baselines) on this dataset. In future work, we plan to expand the 3D HOI dataset and train a more robust evaluator on the larger HOI dataset.
>
> **W2.2 Adding more baseline such as Controllable HOI [2].**
>
> **A2.2:** Please kindly refer to the FAQ1 for a more detailed response. We are willing to discuss it with you if necessary.
>
> **Q1: Feature extractor used in the evaluation.**
>
> **A1:** We adopted a pre-trained movement encoder from HumanML3D [1] and fine-tuned the feature extractor for text-motion-matching on BEHAVE dataset.
>
> **Q2: Is the affordance in APDM estimated for the whole HOI sequence or for every single frame?**
>
> **A2:** The affordance in APDM is predicted for the entire HOI sequence. Currently, we do not account for changes in HOI contact over time. Addressing the challenging problem of different interaction stages will be a focus of our future work.
>
> **Q3: In the affordance estimation stage, you only consider eight body joints for contact. Is this enough for modeling the HOI process? For example, one may kneel on the yoga ball with his knees, which are not included.**
>
> **A3:** Due to the limited size of the current HOI dataset and the lack of interaction diversity, we have focused on eight core contact joints, which are sufficient for modeling typical HOI processes. In the future, we plan to develop a larger dataset with a broader range of interactions (potentially involving the scenario mentioned by the reviewer). This will allow the affordance model to become more detailed, enabling the consideration of additional body joints.
>
> **Q4: Does the model generate constant motion length (as mentioned in L335)?**
>
> **A4:** Yes. We follow Guo et al. [1] to adopt the constant motion length.
>
>
> *[1] Generating Diverse and Natural 3D Human Motions from Text. Guo et al.*
> *[2] Controllable Human-Object Interaction Synthesis. Jiaman Li et al.*

---

> > ### Author Response · Authors · 2024-11-26
> > **Response of Authors**
> >
> > We sincerely thank the reviewer for their valuable feedback on our paper. We have provided a detailed response and uploaded a revised version on OpenReview. With the discussion phase concluding on Dec 2, we would like to confirm if all concerns have been adequately addressed. If so, we would greatly appreciate it if the reviewer could consider updating their score.
> >
> > Thank you for your time and effort!
> > Authors of HOI-Diff

---

> > ### Comment · Reviewer_EXrp · 2024-11-28
> >
> > Thanks to the authors for the response. I incline towards keeping my original rating. Most of my questions have been resolved except the one about HOI evaluation metrics. Why HOI could be evaluated by human-motion-only FID/R-precision still confuses me.

---

> ### Author Response · Authors · 2024-11-27
> **Awaiting Your Response**
>
> Dear Reviewer `EXrp`,
>
> Thank you once again for your time and effort in reviewing our work. With only six days remaining in the discussion phase, we kindly request your feedback on our response. If any part of our explanation remains unclear, please do not hesitate to let us know.
>
> We would greatly appreciate it if you could confirm whether your concerns have been fully addressed. If the issues are resolved, we would be grateful if you could consider reevaluating our work. Should you require any further clarification, we would be happy to provide it promptly before the discussion deadline.
>
> Best,
> Authors of HOI-Diff

---

> ### Author Response · Authors · 2024-11-28
> **Response of Authors**
>
> We thank the reviewer `EXrp` for their additional feedback and insights.
>
> **Why HOI could be evaluated by human-motion-only FID/R-precision?**
>
> We would like to emphasize that our evaluation of HOI goes beyond human-motion metrics to **include interaction quality**, assessed using metrics such as contact distance, penetration score, and L2 distance/Average Precision (AP) for affordance prediction. Furthermore, we conduct a **user study** to further validate the effectiveness of our approach. By combining these metrics, we aim to provide a comprehensive and fair evaluation of HOI performance, consistent with previous and concurrent works [1,2,3,4].
>
> We also wish to address some related questions the reviewer might have:
>
> **1. Why report human-motion-only FID/R-precision in HOI tasks?**
>
> Natural and realistic human motion is a vital aspect of HOI tasks, playing a key role in determining the quality of interactions. In contrast, object motion exhibits limited semantic diversity compared to human poses, as object interactions are inherently tied to human dynamics—for example, a box moves in sync with a person's hands during a holding action. Therefore, evaluating human motion becomes a more critical factor in assessing HOI performance. Furthermore, the use of human-motion-specific metrics, such as FID and R-precision, is a widely recognized practice in many HOI studies [1,2,3,4].
>
>
> **2. Why not train an HOI encoder that can jointly learn human motion, human shape, object motion, and object shape, and then calculate HOI FID/R-precision?**
>
> The current HOI dataset is significantly smaller than available human motion datasets. Training an HOI encoder on such a limited dataset would likely result in overfitting, making it impractical to achieve reliable results.
>
> We hope these clarifications address the reviewer's concerns and provide a deeper understanding of our evaluation.
>
> *[1] CG-HOI: Contact-Guided 3D Human-Object Interaction Generation. Christian Diller et al.
> [2] InterDreamer: Zero-Shot Text to 3D Dynamic Human-Object Interaction. Sirui Xu et al.
> [3] HOIAnimator: Generating Text-prompt Human-object Animations using Novel Perceptive Diffusion Models. Wenfeng Song et al.
> [4] Controllable Human-Object Interaction Synthesis. Jiaman Li et al.*

---

### Official Review · Reviewer_eW1a · 2024-11-02

**Soundness:** 2
**Presentation:** 2
**Contribution:** 2
**Rating:** 5
**Confidence:** 4

**Summary:**

The paper focuses on generating realistic 3D human-object interactions (HOIs) driven by textual prompts using a dual-branch diffusion model (DBDM) and an affordance prediction diffusion model (APDM). It introduces a novel method to estimate contact points and ensure accurate interactions between humans and objects, enhancing the realism and diversity of the generated interactions. The approach is tested on the BEHAVE and OMOMO datasets, demonstrating its effectiveness in handling dynamic objects and generating physically plausible interactions that align closely with textual descriptions.

**Strengths:**

The use of a dual-branch diffusion model combined with an affordance prediction model allows for accurate and diverse generation of human-object interactions, addressing the complexity of such tasks effectively.

The model demonstrates superior performance on standard metrics against baseline models, particularly in generating interactions with unseen objects, which showcases its robustness and ability to generalize across different scenarios.

**Weaknesses:**

The model's performance heavily relies on the quality of datasets like BEHAVE and OMOMO. If these datasets are limited or biased towards specific types of interactions, the model's generalization ability may be compromised. I request authors to test the model with completely unseen objects not included in the BEHAVE or OMOMO datasets. For example, using a text-to-3D related model [1] to create a new object and showing inference results for that object.
[1] DreamFusion: Text-to-3D using 2D Diffusion, ICLR 2023

The qualitative results show that, while the output aligns with text better than the baselines compared with, the generated result can still be physically implausible despite the introduction of Affordance Prediction Diffusion Model, for eg, objects can be seen floating/being carried without enough contact to hold them in place. Also significant penetration is visible between the body and object, but to a much lesser extent than the methods compared against. I'd like to ask authors if they have any plan to add a loss function (e.g. contact loss) especially designed to this issue. Also, I think more sophisticated metric for measuring the penetration, such as "Intersection Volume," needs to be included to properly check the "physical plausibility of the motions". The measure is basically calculated for penetrated volumes; for details on relevant metrics, consider referring to the paper: Physics-aware Hand-object Interaction Denoising. CVPR 2024.

**Questions:**

None

---

> ### Author Response · Authors · 2024-11-21
> **Rebuttal by Authors**
>
> We thank Reviewer eW1a for your comments and constructive suggestions. We respond below to your comments and questions.
>
> **W1: The model's generalization ability.**
>
> **A1:**  We used the train/test split of the OMOMO dataset to evaluate the model's inference capacity on unseen objects, including the small table, white chair, suitcase, and tripod. Notably, the entire test set of the OMOMO dataset consists of unseen objects that are not available during training. The quantitative and qualitative results are presented in Table 1 (right) and Figure 12 in the Appendix. Our method consistently outperforms other baselines. We have updated more details about this in the revised version, please see lines 422-427. We also downloaded several objects from Sketchfab, adjusted them to a reasonable scale, and tested them to further verify the generalization capability. Please refer to lines 1004-1007 and Figure 13 in the Appendix for more details.
>
>
> **W2.1: Issue about floating objects and penetration.**
>
> **A2.1:** The 3D HOI generation task is very difficult and existing methods suffer from these artifacts, e.g., InterDiff [1], OMOMO [2], CG-HOI [3], Interdreamer [4] have many cases with penetrations and floating objects as shown in their webpage teasers.
>
> We designed a distance function to reduce severe body penetration after rendering (please see details in lines 744-749 of the Appendix). However, we acknowledge that certain floating or penetrations are not fully guaranteed. Incorporating additional physical constraints could improve contact accuracy and prevent penetration issues. For instance, fitting the human surface mesh using the SMPL model and introducing contact and collision loss are potential approaches. However, these methods impose substantial computational overhead. In future research, we aim to develop more efficient solutions to address penetration issues and generate physically plausible motions while minimizing computational costs.
>
> *[1] InterDiff: Generating 3D Human-Object Interactions with Physics-Informed Diffusion. Sirui Xu et al.*
>
> *[2] Object Motion Guided Human Motion Synthesis.  Jiaman Li et al.*
>
> *[3] CG-HOI: Contact-Guided 3D Human-Object Interaction Generation. Christian Diller et al.*
>
> *[4] InterDreamer: Zero-Shot Text to 3D Dynamic Human-Object Interaction. Sirui Xu et al.*
>
>
>
>
> **W2.2: The metric of penetration ratio.**
>
> **A2.2:** We thank Reviewer eW1a for this valuable suggestion. We have updated Table 1 to include the penetration metrics, highlighted in blue. Please kindly refer to the FAQ2 for a more detailed response. We are willing to discuss it with you if necessary.

---

> > ### Author Response · Authors · 2024-11-26
> > **Response of Authors**
> >
> > We sincerely thank the reviewer for their valuable feedback on our paper. We have provided a detailed response and uploaded a revised version on OpenReview. With the discussion phase concluding on Dec 2, we would like to confirm if all concerns have been adequately addressed. If so, we would greatly appreciate it if the reviewer could consider updating their score.
> >
> > Thank you for your time and effort!
> > Authors of HOI-Diff

---

> > ### Comment · Reviewer_eW1a · 2024-12-01
> > **Thanks for the rebuttal**
> >
> > Thanks for the rebuttal.
> >
> > Thanks for updating the Table 1 reflecting the penetration measure; however I was bit disappointed by the fact that the proposed method is actually less competitive than other methods, in terms of penetration. I think penetration is very important aspect, especially in the HOI generation. However, I do not think the derived method is properly tackling the aspect.
> >
> > Also, thanks for including the unseen generation cases in the results. However, I cannot see that the proposed method outperforms other methods especially in the unseen object cases. Also, I request authors to reason why the proposed method is better in such generalization cases than others.

---

> > > ### Author Response · Authors · 2024-12-02
> > > **Response by Authors**
> > >
> > > We thank the reviewer eW1a for their additional feedback and insights.
> > >
> > > **Q1. Why are the results on contact distance the best, while the penetration score is not optimal?**
> > >
> > > **A1.** A good contact quality requires achieving a balance where there is neither penetration nor floating. For instance, achieving a penetration score of 0 can be artificially accomplished by placing the object far away from the human (e.g., 10 meters); however, this fails to reflect meaningful contact quality. This limitation indicates that penetration score alone cannot fully capture the realism of interactions.
> > >
> > > From our experiments, we observed that many baselines, such as InterDiff, MDM* and PriorMDM*, often produce results where objects float significantly away from the human, resulting in artificially low penetration scores. This issue is similarly noted in Table 1 of Controllable HOI [1] and Table 2 of OMOMO [2], further underscoring the need for a more comprehensive evaluation metric. (This trend is also visible in the visualizations provided in Figure 3 of Controllable HOI[1].)
> > >
> > > To address this, we propose combining Contact Distance and Penetration Score into a unified HOI score, which offers a more robust and accurate framework for evaluating interaction realism. Specifically, since both metrics are measured in centimeters (cm), we compute the final score by directly summing the two values. As shown in the Table below, our method achieves the best results, demonstrating that our HOI approach captures more realistic and meaningful interactions.
> > >
> > >
> > > **Table 1. Quantitative results in the metric of HOI score on two datasets. The lower is better.**
> > > | | MDM* | PriorMDM* | InterDiff | Chois  | Ours |
> > > | -------- | ------- | -------- | ------- | --------     | ------- |
> > > | BEHAVE | 0.968 | 0.986 | 0.926 | -         | **0.857**  |
> > > | OMOMO | 1.178 | 0.903 | 1.226 | 0.802 | **0.716** |
> > >
> > > [1] Controllable Human-Object Interaction Synthesis. Jiaman Li et al.
> > >
> > > [2] Object Motion Guided Human Motion Synthesis. Jiaman Li et al.
> > >
> > >
> > > **Q2. I cannot see that the proposed method outperforms other methods especially in the unseen object case.**
> > > **A2.** Due to the expiration of the PDF revision deadline, we provide supplementary comparative visualizations on unseen objects via this anonymous [link](https://anonymousrepo2025.github.io/). As shown, our method generates more plausible and realistic HOIs, demonstrating its effectiveness in handling unseen objects.
> > >
> > > MDM* [1] and PriorMDM* [2] do not learn good spatial relationships between people and objects by themselves, and thus generalize even less well for unseen objects. Interdiff [3] is also not inherently designed for text-driven, so it doesn't generate good results either. When Chois [4] removes the dependency on object trajectories, the performance drops dramatically.  Moreover, Chois [4] can't only learn two-handed interactions, and can't control other body part interactions through text. All of the above baselines only implicitly learn the spatial relationship between humans and objects, and cannot fully capture accurate contact information. However, our approach leverages pretrained priors to generate smooth human motions, which can exhibit strong generalization capabilities. Besides, we specifically designed an affordance prediction model that estimates contact positions based on the geometry of the object, ensuring more realistic and plausible interactions, which can further correct the HOI contact, even for the unseen objects.
> > >
> > > *[1] Human Motion Diffusion Model. Guy Tevet et al.
> > > [2] PriorMDM: Human Motion Diffusion as a Generative Prior. Yonatan Shafir et al.
> > > [3] InterDiff: Generating 3D Human-Object Interactions with Physics-Informed Diffusion. Sirui Xu et al.
> > > [4] Controllable Human-Object Interaction Synthesis. Jiaman Li et al.*

---

> > > > ### Comment · Reviewer_eW1a · 2024-12-02
> > > >
> > > > Thanks for clarifying the points.
> > > >
> > > > Especially, I become confident about the penetration measure thanks to the explanation on the trade-offs on penetration and contact distance.
> > > >
> > > > However, I cannot easily follow the argument here yet, "our approach leverages pretrained priors to generate smooth human motions, which can exhibit strong generalization capabilities". Especially, I think it may require thorough ablation studies for `Ours w/o pretrain' in unseen settings.

---

> ### Author Response · Authors · 2024-12-03
> **Response by Authors**
>
> We thank Reviewer eW1a for the additional feedback and are pleased to hear that the HOI metrics have been clearly presented.
>
> **Q1. Why pretrained priors can improve the generalization and more ablation studies for `Ours w/o pretrain' in unseen settings.**
> **A1.** Since the current HOI dataset is relatively small, we leverage pretrained priors to fine-tune on the HOI dataset. These priors allow us to generate more consistent and realistic human body motions, even for unseen objects described in the text. Additionally, this fine-tuning strategy minimizes the impact on the object branch, enabling it to better learn object poses simultaneously.
>
> Supplementary ablation comparisons are provided in Figure 17 of the anonymous [**link**](https://anonymousrepo2025.github.io/). As shown, without pretraining, the human body's motion range is limited and inconsistent with text description, and objects often exhibit unnatural poses.
>
> The quantitative ablation results on OMOMO's unseen test set, presented in Table 2 of the main paper, further demonstrate that the performance of the model without pretraining (w/o pretrain in the third row) is inferior to both our approach without interaction correction and our full method with interaction correction.
>
> We look forward to your response and hope that, if all concerns have been addressed, the reviewer will find the manuscript deserving of an acceptance/higher score.

---

### Official Review · Reviewer_1W4D · 2024-11-04

**Soundness:** 2
**Presentation:** 2
**Contribution:** 2
**Rating:** 5
**Confidence:** 5

**Summary:**

This paper presents a text-driven human-object interaction generation method with a modular design. The proposed method includes three steps, a dual-branch diffusion, an affordance prediction diffusion model, and the refinement stage. Firstly, the dual-branch generates a coarse human-object motion. Then the affordance prediction diffusion model predicts the contacting points. Finally, the refinement stage refines the coarse human-object motion by the contacting points. In the experimental section, the proposed method is evaluated on public datasets, BEHAVE and OMOMO.

**Strengths:**

Here I highlight the strengths of the paper:

1. The modular design that starts with a coarse grain and refines to a fine grain is valuable for human-object interaction generation.
2. The ablation study is sufficient to prove the effectiveness of the proposed method.
3. The presentation of this paper is clear, especially the pipeline of the proposed method.

**Weaknesses:**

Although the coarse-to-fine modular design is valuable, this paper only considers the contact between body parts and objects. This method neglects the important contact between hand and object in human-object interaction. Here, I highlight my concerns.

1. Lack of citations and discussion of related work. Most of the methods cited and compared are those before 2023. There are a lot of related works that appear in 2024, such as [1,2,...]. These new works should be discussed and compared.

[1]CG-HOI: Contact-Guided 3D Human-Object Interaction Generation
[2]HOIAnimator: Generating Text-prompt Human-object Animations using Novel Perceptive Diffusion Models
[3]InterFusion: Text-Driven Generation of 3D Human-Object Interaction
[4]Controllable Human-Object Interaction Synthesis
[5]F-HOI: Toward Fine-grained Semantic-Aligned 3D Human-Object Interactions

2. The hand-object interaction modeling is neglected. The proposed method only models the contacting points between body parts and the object, which is very coarse-grained for human-object interaction. This causes the method to generate effect distortions in the contact area between hand and object.

3. The compared methods are almost all human motion generation methods; only InterDiff is a human-object interaction generation method. There is a lack of comparison of human-object interaction generation methods, making it difficult to evaluate the effect of the proposed method in this domain.

**Questions:**

1. In the Line 256~257, why consider these 8 body joints? And which eight? Only five are listed in the paper.

2. In the experiment section, most of the metrics are general human motion generation metrics, only the contact distance is for human-object generation. Why not use the metrics of InterDiff for human-object generation, such as MPJPE-H, Trans. Err, Rot. Err, MPJPE-O. Otherwise, it is difficult to measure the effect of the method on human-object generation.

3. Why are the hands barely moving? They look flat.

---

> ### Author Response · Authors · 2024-11-21
> **Rebuttal by Authors**
>
> We thank Reviewer 1W4D for your comments and constructive suggestions. We respond below to your comments and questions.
>
> **W1:Lack of citations and discussion of related work?**
>
> **A1:** We thank Reviewer 1W4D for this suggestion. We have cited and discussed the relevant works in the related work section of the revised PDF (see lines 124-132). Regarding experimental comparison, please refer to FAQ2 for a more detailed response.
>
> **W2: The hand-object interaction modeling is neglected.?**
>
> **A2:** Existing HOI datasets mainly focus on whole-body interactions with large-scale movements and diverse interactions but no detailed hand motions. On the other hand, hand-object datasets provide detailed hand information but are limited to small objects and grasp-based interactions. This work was initially motivated by the goal of addressing more diverse interactions, encompassing various interaction parts and actions. However, due to the limitations of the current datasets, it is not yet feasible to fully solve whole-body HOI generation for large-scale movements with related hand interactions. Thus we didn’t synthesize the fine-grained hand motion, same as many other concurrent works [1,2,3,4].
>
>
> **W3: There is a lack of comparison of human-object interaction generation methods**
>
> **A3:** We have supplemented the comparison with Choi [4] in the revised Table 1 of the main paper. We elaborate on more details in the response of FAQ1.
>
>
> **Q1: Why consider these 8 body joints? And which eight? Only five are listed in the paper**
>
> **A1:** Given the limited size of the current HOI datasets and the lack of diverse interactions, we have chosen to focus on 8 core contact joints to address common interaction activities. These joints include the pelvis, neck, both feet (left and right), both shoulders (left and right), and both hands (left and right).  In the future, we plan to develop a larger dataset, enabling the affordance model to become more fine-grained and capable of supporting a greater variety of body parts.
>
> **Q2: Why not use the metrics of InterDiff for human-object generation**
>
> **A2:** We thank Reviewer 1W4D for this valuable suggestion. However, we would like to clarify that the metrics used in InterDiff are designed for deterministic prediction task, which are commonly applied in human motion prediction or reconstruction rather than text-driven motion generation. In text-driven motion generation, the outputs may exhibit diversity, meaning they may not align perfectly with the ground truth at each time step or spatially.
>
> As also suggested by other reviewers about supplementing more metrics for HOI generation, we have updated Table 1 in the main paper to include the penetration metrics, highlighted in blue. Please refer to the FAQ2 (additional metric of penetration score) for more details.
>
>
> **Q3: Why are the hands barely moving? They look flat.**
>
> **A3:**  As mentioned in the response to **W2**, the dataset used for training does not include detailed hand parameters, such as those provided by MANO. Consequently, we can only pad zeros for hand parameters and fit the skeleton joints to the SMPL mesh, resulting in visually flat hands.
>
> *[1] CG-HOI: Contact-Guided 3D Human-Object Interaction Generation. Christian Diller et al.*
>
> *[2] InterDreamer: Zero-Shot Text to 3D Dynamic Human-Object Interaction. Sirui Xu et al.*
>
> *[3] HOIAnimator: Generating Text-prompt Human-object Animations using Novel Perceptive Diffusion Models. Wenfeng Song et al.*
>
> *[4] Controllable Human-Object Interaction Synthesis. Jiaman Li et al.*

---

> > ### Comment · Reviewer_1W4D · 2024-11-23
> >
> > Thanks to the authors for their work on the rebuttal. Two concerns remain, however.
> >
> > W2A2: About the modeling of hand-object interaction, the eight contact body points are coarse-grained. Compared with this paper, HOIAnimator models the interaction by an interaction contact field, which is more meticulously.
> >
> > W3A3: Why are the results on contact distance best, while the penetration score is not the best?

---

> ### Author Response · Authors · 2024-11-27
> **Awaiting Your Response**
>
> Dear Reviewer `1W4D`,
>
> Thank you once again for your time and effort in reviewing our work. With only six days remaining in the discussion phase, we kindly request your feedback on our response. If any part of our explanation remains unclear, please do not hesitate to let us know.
>
> We would greatly appreciate it if you could confirm whether your concerns have been fully addressed. If the issues are resolved, we would be grateful if you could consider reevaluating our work. Should you require any further clarification, we would be happy to provide it promptly before the discussion deadline.
>
> Best,
> Authors of HOI-Diff

---

> ### Author Response · Authors · 2024-12-02
> **Awaiting Your Response (2 Days Left)**
>
> Dear Reviewer `1W4D`,
>
> Thank you once again for your time and effort in reviewing our work. We sincerely appreciate your constructive feedback.
>
> We kindly request your confirmation on whether the concerns you raised have been fully addressed in the revised submission. If the issues are resolved to your satisfaction, we would be deeply grateful if you could consider reevaluating our work. Should you need any further clarification or additional information, we are more than happy to provide it promptly before the discussion deadline.
>
>
> Best regards,
> Authors of HOI-Diff

---

### Official Review · Reviewer_WtJm · 2024-11-04

**Soundness:** 3
**Presentation:** 3
**Contribution:** 3
**Rating:** 5
**Confidence:** 4

**Summary:**

This paper address the problem of text-driven 3D human-object interaction synthesis. The task is decomposed into simpler sub-tasks, i.e., human and object motion generation, affordance estimation and affordance-guided interaction correction. A dual-branch diffusion model (DBDM) is developed to generate the coarse human and object motion. Then, an affordance prediction diffusion model is designed to predict the contacting area between the human and object. Finally, the estimated contacting points are incorporated into the classifier-guidance to achieve accurate and close contact between humans and objects. Experiments on BEHAVE and OMOMO show realistic HOI generation results.

**Strengths:**

1. The topic of text-driven HOI synthesis is of large importance in the community;
2. The annotation on BEHAVE is beneficial for the community;
3. Experimental results show the effectiveness of the proposed method;
4. The paper is well written and organized.

**Weaknesses:**

1. The motivation of decomposing HOI synthesis into simpler sub-tasks is not clearly elaborated, which should be discussed. Why the model generate a roughly coherent HOI motion at first rather than directly generate a high quality HOI motion?
2. The authors claim that the text-driven 3D HOI synthesis with a diverse set of interactions is under-explored and existing datasets lack either HOIs or textual descriptions. However, many works have been proposed for this task such as CG-HOI and CHOIS. They also annotate the BEHAVE dataset with textual descriptions or build 3D HOI dataset with textual descriptions, such as OMOMO. The claim should be modified in the introduction section;
3. The authors claim that the model can produce physically plausible results, yet no physical constraints are integrated to guarantee that. And based on the visualization results, the physical constraint cannot be kept, mainly due to the dataset.

**Questions:**

My main concerns lie in the motivation of the paper, the claim of the introduction and the physical constraints of the generated results. If the concerns are well addressed, I am willing to raise my rating.

---

> ### Author Response · Authors · 2024-11-27
> **Awaiting Your Response**
>
> Dear Reviewer `WtJm`,
>
> Thank you once again for your time and effort in reviewing our work. With only six days remaining in the discussion phase, we kindly request your feedback on our response. If any part of our explanation remains unclear, please do not hesitate to let us know.
>
> We would greatly appreciate it if you could confirm whether your concerns have been fully addressed. If the issues are resolved, we would be grateful if you could consider reevaluating our work. Should you require any further clarification, we would be happy to provide it promptly before the discussion deadline.
>
> Best,
> Authors of HOI-Diff

---

> > ### Author Response · Authors · 2024-12-02
> > **Awaiting Your Response (2 Days Left)**
> >
> > Dear Reviewer `WtJm`,
> >
> > Thank you once again for your time and effort in reviewing our work. We sincerely appreciate your constructive feedback.
> >
> > We kindly request your confirmation on whether the concerns you raised have been fully addressed in the revised submission. If the issues are resolved to your satisfaction, we would be deeply grateful if you could consider reevaluating our work. Should you need any further clarification or additional information, we are more than happy to provide it promptly before the discussion deadline.
> >
> >
> > Best regards,
> > Authors of HOI-Diff

---

### Author Response · Authors · 2024-11-21
**Author Rebuttal by Authors**

We sincerely thank all the reviewers for providing constructive feedback that helped us improve the paper. We are glad the reviewers find that
- Our work is effective and valuable (Reviewer WtJm, 1W4D, eW1a, EXrp)
- Our presentation is clear and well-motivated  (Reviewer WtJm, 1W4D, EXrp)
- Our performance is outstanding (Reviewer WtJm, 1W4D, eW1a, EXrp)


We also list here the responses to a few frequently asked questions for reviewers to refer to.

### **FAQ1: Compare with more HOI baselines, such as Controllable HOI [4].  ---1W4D, eW1a**

**A1:** We thank the reviewers’ suggestions for adding more baselines. However, most related works, such as CG-HOI [1], Interdreamer [2], HOIAnimator [3], F-HOI [3], and Interfusion [4], have not released their code. Additionally, F-HOI [3] and Interfusion [4] are specifically designed for **static** 3D HOI generation rather than full HOI sequences.


Except for these works, only Controllable HOI (Chois) [4] have made their code available. However, this method operates under different task settings, relying on initial states and object waypoints as conditions. Furthermore, it is limited to handling two-handed interactions, such as moving a cloth stand with one hand. To ensure a fair comparison, we modified their framework by removing dependencies on initial states and object waypoints, and we conducted experiments exclusively on the OMOMO dataset for two-hand interaction. As shown in the 6th row of Table 1 in the main paper, the performance of Chois remains suboptimal, likely due to the absence of object waypoint information. For your convenience, we also list the results below for easier reference.


**Table 1. Quantitative comparison on OMOMO dataset between Chois and Ours. (*Pene.: penetration score*)**
| | FID↓ | R-precision↑ | Diversity→ | Contact Distance↓ | Pene.↓ | Foot Skate Ratio↓ |
| -------- | ------- | -------- | ------- |------- |------- |------- |
| Real |0.57 | 0.63 | 9.98 | - | - |
| Chois | 9.69 | 0.24 | 7.33 | 0.432 | 0.37 | 0.165|
| Ours |  8.76 | 0.31 | 8.13 | 0.326 | 0.39| 0.141 |




### **FAQ2: Supplement additional HOI metrics, such as penetration score. ---1W4D, eW1a, EXrp**


**A2:** We thank reviewers for pointing this out. We have updated Table 1 to include the penetration metrics, highlighted in blue. For your convenience, we also list the results below for easier reference. Specifically, we followed the method outlined by Li et al. [4] to compute the penetration score. We first use the SMPL model to recover the human body mesh. Then each vertex of the human body was queried against the precomputed Signed Distance Field (SDF) of the object. The penetration score was then derived by averaging the negative distance values, which represent the extent of penetration, measured in centimeters. More details are supplemented in section D of Appendix, highlighted in blue color. Notably, some methods, such as InterDiff, achieve lower penetration scores primarily due to severe object-floating issues.


**Table 2. Quantitative results in the metric of penetration score on two datasets**
| | MDM* | PriorMDM* | InterDiff | Chois | Ours |
| -------- | ------- | -------- | ------- | -------- | ------- |
| BEHAVE | 0.52 | 0.57 | 0.42 | - | 0.51 |
| OMOMO | 0.41 | 0.38 | 0.32 | 0.37 | 0.39 |


*[1] CG-HOI: Contact-Guided 3D Human-Object Interaction Generation. Christian Diller et al.*


*[2] InterDreamer: Zero-Shot Text to 3D Dynamic Human-Object Interaction. Sirui Xu et al.*


*[3] HOIAnimator: Generating Text-prompt Human-object Animations using Novel Perceptive Diffusion Models. Wenfeng Song et al.*


*[4] Controllable Human-Object Interaction Synthesis. Jiaman Li et al.*

---

### Meta-Review · Area_Chair_Lo3o · 2024-12-21

**Metareview:**

This paper proposes a method for synthesizing 3D hand-object interactions from text.

The reviewers appreciate the good results as well as extension of existing datasets with text annotations.

Several weaknesses and questions were raised, concerning the motivation, clarity and experimentation.  There are several questions regarding the evaluation metrics and lack of comparisons to qualified baselines.  These points are somewhat addressed in the author responses, but not in a very convincing manner.

The final scores are 3 borderline reject and 1 borderline accept (reviewer EXrp).  Reviewer EXrp also states explicitly a lack of support for the paper.  After reading the paper, review and author responses, the AC recommends that the paper be rejected.  Authors are recommended to revise their papers based on the reviewer questions and responses.  Furthermore, they are recommended to adjust their experiments and evaluation measures to further demonstrate the effectiveness of their approach and improvements of existing works.

**Additional Comments On Reviewer Discussion:**

During the closed discussion, reviewer EXrp (who gave a borderline accept) mentioned that after reading the reviews and responses by the authors, they are neutral and do not have motivation to defend the work.  The other eW1a does not adjust their score.

Unfortunately two of the other reviewers are not responsive and the AC is using qualified judgement to evaluate the author responses.  Some points, especially regarding (lack) of experimental baselines, are not convincing.  The purpose of experimental baselines is to where possible, compare against state-of-the-art.  However, if state-of-the-art are concurrent works and or do not provide source codes, it is the onus of the authors to provide alternative baselines (even if not directly comparable) to convince readers on the effectiveness of their approach.

---

### Decision · Program_Chairs · 2025-01-22

Reject